## nature
# ecology & evolution

# ARTICLES
# Vision and vocal communication guide three-dimensional spatial coordination of zebra finches during wind-tunnel flights

Fabian Arnold [1,2,4], Michael S. Staniszewski [1,2,5], Lisa Pelzl[1,2,6], Claudia Ramenda[1,3], Manfred Gahr[1,3] and Susanne Hoffmann [1,3 ✉]

**Animal collective motion is a natural phenomenon readily observable in various taxa. Although theoretical models can predict the macroscopic pattern of group movements based on the relative spatial position of group members, it is poorly understood how group members exchange directional information, which enables the spatial coordination between individuals during collective motion. To test if vocalizations emitted during flocking flight are used by birds to transmit directional information between group members, we recorded vocal behaviour, head orientation and spatial position of each individual in a small flock of zebra finches (*Taeniopygia guttata*) flying in a wind tunnel. We found that the finches can use both visual and acoustic cues for three-dimensional flock coordination. When visual information is insufficient, birds can increasingly exploit active vocal communication to avoid collisions with flock mates. Our study furthers the mechanistic understanding of collective motion in birds and highlights the impact interindividual vocal interactions can have on group performances in these animals.**

Many bird species fly in groups, either forming highly structured V-shaped flight formations, which can provide energetic benefits[1,2], or aggregating in dense cluster flocks to reduce predation risk[3,4]. To coordinate movements and hence avoid collisions in moving groups, directional information has to be transferred between group members[5–7]. Theoretical models, which predict the macroscopic pattern of group movements based on the relative spatial position of group members[8–11], often assume that each individual in a group visually observes the behaviour of surrounding group mates to adapt their own movements to those of their neighbours[6,12,13]. Although vision seems to be the primary channel for birds to gather environmental information, the sensory mechanisms by which information transfer is achieved in flocks of birds during flight are still elusive. We hypothesize that besides vision, vocalizations emitted during flight can play a role in spatial coordination of bird flocks. Although, many bird species are known to emit calls during flight, the function of the so-called 'flight calls' is still largely unknown. Anecdotal evidence suggests a potential role of flight calls in group maintenance[14,15].

In their natural habitat, the semi-arid parts of Australia, zebra finches (*Taeniopygia guttata*) form cluster flocks to reduce predation risk while moving between their nesting trees and foraging grounds[16,17]. To understand which sensory modalities zebra finches use to coordinate their three-dimensional (3D) spatial positions during flocking flight, we recorded the individual vocal behaviour, fine-scale head orientation and spatial position of six adult zebra finches while flying together in the flight section of a large wind tunnel (Fig. 1a). We show that during flocking flight zebra finches not only visually observe their environment, but can also vocally interact with each other to avoid collisions within the flying flock.

## Results

**Dynamic in-flight flock organization.** It is commonly assumed that during flocking, flock members follow three basic interaction rules: Attraction, Repulsion and Alignment, to coordinate spatial positions between each other[18]. To study the spatial organization of our zebra finch flock during flight, the spatial positions of all birds in the flight section were tracked in every fifth frame (sample rate: 24 Hz (that is, frames per second)) of the synchronized footage recorded by two high-speed digital video cameras (Camera 1: centred upwind view, Fig. 1a,b; Camera 2: upturned vertical view, Fig. 1a,c) for the entire duration (51.7, 58.3, 69.2 and 127 s) of four (session 2, 5, 8 and 13) out of 13 flight sessions. Flight paths were reconstructed from the tracking data for each bird in the flock, with horizontal and vertical coordinates delivered by Camera 1 and coordinates in wind direction delivered by Camera 2. The data show that each bird mainly occupied a particular area in the flight section, and that this spatial preference was stable over different flight sessions. Bird Green, for example, was preferentially flying very low above the flight section's floor, and bird Lilac preferred to fly at upwind positions in front of the flock (Fig. 1d, Extended Data Figs. 1 and 3 and Supplementary Information).

Despite their preference in flight area, all birds constantly changed their spatial positions fast and rhythmically along the horizontal dimension of the flight section (Fig. 1e–g, Extended Data Figs. 2 and 4, Supplementary Video 1 and Supplementary Information). This behaviour is reminiscent of the flight behaviour of wild zebra finches: when being surprised in flight by a predator, zebra finches fly in a rapid zig-zag course low above the ground, heading for nearby vegetation[16]. Whether the sideways oscillating flight manoeuvres, which are performed by both wild birds in open

[1]Department of Behavioural Neurobiology, Max Planck Institute for Ornithology, Seewiesen, Germany. [2]Faculty of Biology, Ludwig-Maximilians-University of Munich, Planegg-Martinsried, Germany. [3] Department of Behavioural Neurobiology, Max Planck Institute for Biological Intelligence (in Foundation), Seewiesen, Germany. [4]Present address: TUM School of Life Sciences, Technical University of Munich, Freising, Germany. [5]Present address: Faculty of Sciences and Bioengineering Sciences, Vrije Universiteit Brussel, Elsene, Belgium. [6]Present address: Faculty of Biology, Ludwig-Maximilians-University of Munich, Planegg-Martinsried, Germany. ✉e-mail: shoffmann@orn.mpg.de

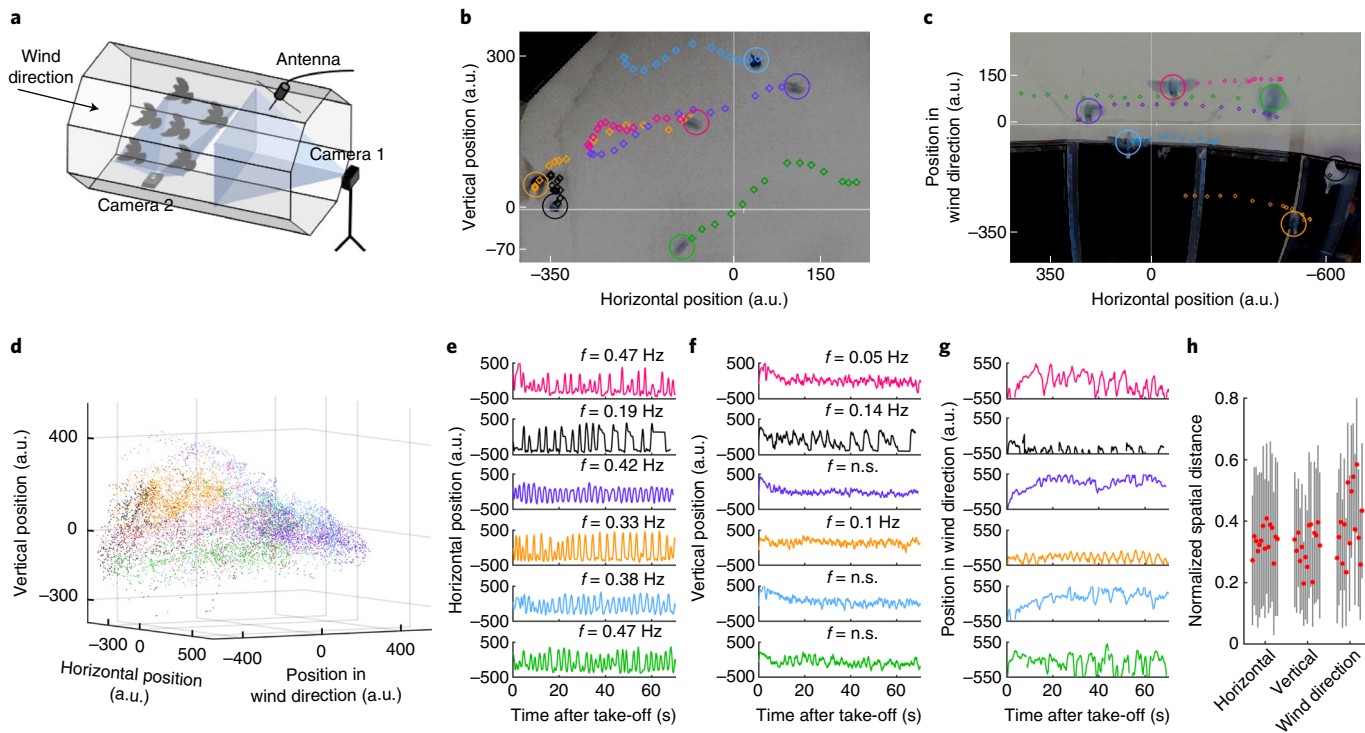

**Fig. 1 | Zebra finches dynamically change their spatial positions within the flying flock. a**, Schematic representation of the experimental setup in the flight section (not to scale). Blue-shaded areas: cameras' fields of view. **b,c**, Tracked positions (sample rate: 24 Hz) of each bird are indicated for a time period of 586 ms in colour-reversed cutouts of freeze frames of the footage taken with Camera 1 (**b**) and Camera 2 (**c**). Coloured circles indicate the birds' positions at the end of the sequence. Note the alignment of movement trajectories of the birds Pink, Lilac and Light blue, and of the birds Orange and Green. **d**, Pseudo 3D representation of all birds' spatial positions (sample rate: 24 Hz) during one example of flight sessions (session 8, duration: 85.4 s), indicating the preferred area in the flight section occupied by each bird. **e–g**, Reconstructed flight paths (sample rate: 24 Hz) in the horizontal (**e**) and vertical (**f**) dimension, and in wind direction (**g**), for each bird during the flight session shown in **d**. *f*, frequency of position change (Hz = cycles per second); n.s., not significant. Note the rhythmic fluctuations of flight paths in the horizontal dimension. Axis scaling in **b** to **g**: negative values = bottom, left and downwind positions in the flight section. Marker colours in **b** to **g** correspond to the birds' IDs. **h**, Mean (red dots) and s.d. (grey lines) of spatial distances normalized to the maximum distance detected for each bird pairing are shown for *n* = 15 bird pairings in the horizontal and vertical dimension, and in wind direction for the flight session shown in **d**.

space and domesticated birds in the wind tunnel's flight section, are caused by the close proximity to the ground or are part of an escape reaction is yet unknown.

From the tracking data, we further calculated the spatial distances in all three dimensions between all pairwise combinations of birds throughout the four flight sessions (sample rate: 24 Hz). When normalized to the maximum distance detected for each bird pairing, each dimension and each flight session, mean distances of bird pairings in all dimensions were narrowly distributed within a range of 27.7–38.0% of maximum distance (Fig. 1h and Supplementary Table 1). This may indicate that during flocking flight, zebra finches actively balance Attraction and Repulsion to maintain a stable 3D distance towards all other members of the flock. Owing to the spatial limitations in the wind tunnel's flight section, we did not expect the zebra finches to perform large-scale flight manoeuvres with movements aligned between all flock members (Extended Data Fig. 5 and Supplementary Information), as can be observed, for example, in freely flying flocks of homing pigeons (*Columba livia domestica*)[19] and white storks (*Ciconia Ciconia*)[20].

**Visually guided horizontal repositioning.** When observing the dynamic spatial organization of our zebra finch flock, a question immediately arises: how do the birds prevent collisions during their frequent horizontal position changes? When considering the spatial limitation experienced by the flock of six birds during flight

in the flight section and their highly dynamic flight style, collision rates seemed to be astonishingly low (median: 0.02 Hz; interquartile range (IQR): 0–0.03 Hz; *n* = 13 sessions) during flocking flight (in total 16 collisions in 13 min of analysed flight time). In birds, the visual system represents the main input channel for environmental information. To tackle the above question, we therefore first investigated the role of vision during flocking flight, and tested whether a bird's viewing direction was correlated with the direction of horizontal position change. As gaze changes are governed by head movements in birds[21], we used a bird's head direction as an indicator for the orientation of its visual axis. We tracked (sample rate: 120 Hz) the position of a bird's beak tip and neck in each frame of the footage during ten horizontal position changes (Fig. 2a and Supplementary Video 2) per bird, and found a strong interaction between a bird's head angle relative to the wind direction and its direction of horizontal position change. During horizontal position changes, the birds always turned their heads in the direction of the position change (Fig. 2b). While the population's median absolute angle of position change was 84.0° (IQR: 78.6–87.2°; *n* = 60) relative to 0° in wind direction, the population's median absolute head turning angle was 36.0° (IQR: 26.4–42.5°; *n* = 60; see Supplementary Information for results on head movements during solo flight). The eyes of zebra finches are positioned laterally on their heads[22] and each retina features a small region of highest ganglion cell density (fovea, that is, region of highest visual spatial resolution) at an area

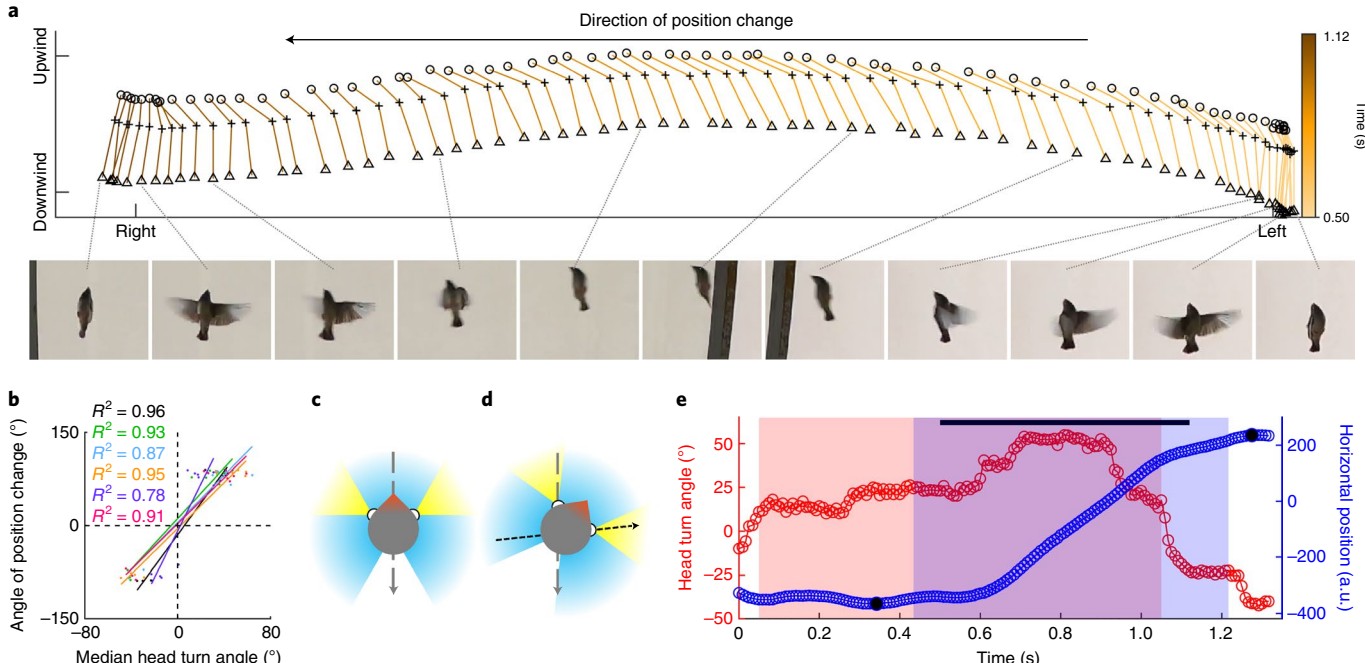

**Fig. 2 | Horizontal position changes are accompanied by head turns. a**, Head and body orientation of bird Orange (ventral view) during one example of position changes to the right, tracked (sample rate: 120 Hz) in the footage of Camera 2. Circles: beak tip positions; plus signs: neck positions; upward pointing triangles: tail base positions. Cutouts of freeze frames of the footage taken with Camera 2 show the bird's head and body posture for 11 time points during the position change. **b**, In all birds, the median angle of head turn during horizontal position change in flocking flight is positively correlated (linear mixed effects model (LMM), estimates ± s.e.m.: 2.05 ± 0.1, $P < 0.001$, $t = 21.0$) with the direction of position change relative to zero degrees in wind direction. Coloured dots: individual data points; coloured lines: fitted linear regression models ($R^2$ values are indicated for each bird); colours: bird IDs. Negative values: left-hand positions in the flight section. $n = 10$ horizontal position changes per bird. **c,d**, Schematic representation of a bird head's orientation (dorsal view) during straight flight (**c**) and during a horizontal position change to the right (**d**). The overall visual field of a zebra finch[22] and spatial areas with high visual acuity[23] are indicated in blue and yellow, respectively. Black arrow: direction of position change; grey arrow: wind direction. **e**, Head turn angles (red circles) and horizontal positions (blue circles) of bird Orange during the horizontal position change shown in **a**. Light red area: time period of significant head turn; light blue area: time period of significant position change; purple area: product of the overlap of the light red and blue areas; black line: time period shown in **a**; black dots: most lateral positions.

that receives visual input from horizontal positions at 60° relative to the midsagittal plane[23]. By turning their heads by about 36° during horizontal position changes, the zebra finches roughly align the foveal area in the retina of one eye with their direction of position change, and in the retina of the other eye with the wind direction (Fig. 2c,d). Thus, head turns in the direction of position change may indicate that the birds use visual cues while repositioning themselves within the flock. This hypothesis is supported by a study on zebra finch head movements performed during an obstacle avoidance task. In this study, instead of fixating on the obstacle, zebra finches turned their head in the direction of movement while navigating around the obstacle[24].

Interestingly, birds usually turned their heads already before they initiated the position change (Fig. 2e). The delay between initiation of head turn and initiation of position change was variable within and between individuals, ranging from −16.6 ms (position change preceding head turn) to 736.6 ms (position change following head turn). A population median delay of 215.9 ms ($n = 60$ position changes) may provide sufficient time for a bird to visually evaluate whether the flight path is clear before initiating a horizontal movement. The large variability in the delay between head turn onset and position change onset opposes the hypothesis that head turning behaviour may only be a motoric byproduct of the position change, and may be needed to steer the bird's body in the direction of position change.

Theoretical models that incorporate visual input to predict the pattern of collective movements generally assume fixed values for an individual's visual field and consequently for the spatial area in which the individual is able to perceive visual information from conspecifics[13]. Our data demonstrate that a bird's visual field during flight is not static, but visual range can be increased by head movements. A theoretical increase in perceptual range has been shown to affect the output of collective behaviour models[13]. Incorporating natural dynamics of visual ranges in these models may therefore result in even more realistic predictions of collective behavioural patterns.

**In-flight vocal behaviour.** Zebra finches are highly vocal birds, emitting thousands of communication calls per day[25]. Two of the most frequent call types, the distance call and the stack call, are also uttered during flight[16]. While it has been suggested that distance calls are used to localize conspecifics, stack calls seem to convey information about a bird's intention to execute a certain movement[16,26]. Based on vocal signatures unique to each bird and each call type, calls can be used for individual recognition[27]. To observe the individual vocal behaviour of birds flying in the wind tunnel, we equipped each zebra finch with a light-weight radio-telemetric microphone transmitter[28–30]. During flocking flight sessions, the rate of vocal emissions in our zebra finches was generally low (median: 0.03 Hz; IQR: 0.02–0.07 Hz; $n = 65$ (13 sessions, 5 birds)). While most vocalizations were emitted during the first four seconds following take-off from the perch, vocalization count plateaued at a low level during the subsequent phase of sustained flight (Fig. 3a, top panel). We suggest that zebra finches lower their vocalization rate during flight to reduce predation risk. A high call rate in free

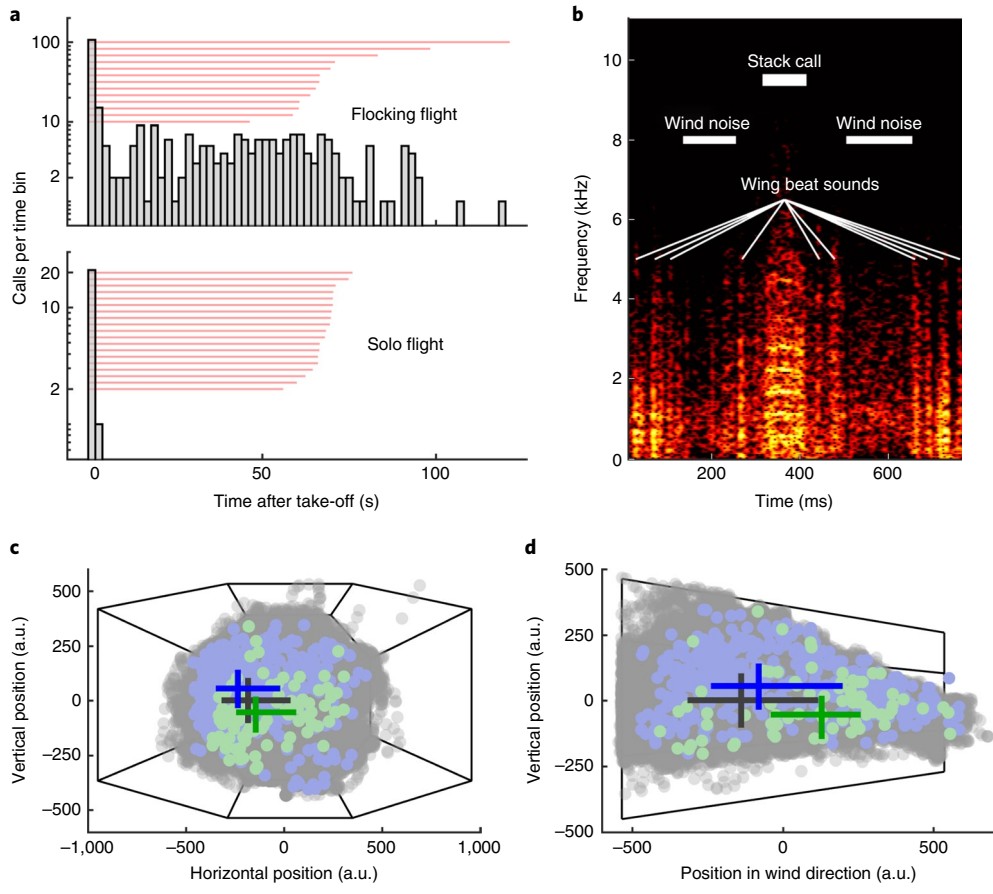

**Fig. 3 | Vocal activity during flight depends on social context. a,** Calling activity differs between flocking and solo flight sessions. Histogram bin size: 2 ms. Red lines: duration of flight sessions. $n = 308$ calls emitted by 5 birds in 13 flocking flight sessions and $n = 22$ calls emitted by 1 bird in 19 solo flight sessions. **b,** Spectrogram (fast Fourier transform length: 512, 99.6% overlap, Hamming window) of a stack call emitted by bird Green during flocking flight. Light colours represent high energy. **c,d,** When emitting a stack call during flight, the calling bird ($n = 93$; light green dots) was positioned at a significantly (LMM, estimates ± s.e.m.: 252.3 ± 49.9, $P < 0.001$, $t = 5.06$) lower, significantly (LMM, estimates ± s.e.m.: 154.6 ± 74.3, $P = 0.038$, $t = 2.08$) more right (**c**) and significantly (LMM, estimates ± s.e.m.: 564.8 ± 97.2, $P < 0.001$, $t = 5.81$) further upwind (**d**) position than its flock mates ($n = 465$; light blue dots). Grey dots: all positions of all birds during four flight sessions; dark blue, dark green and dark grey lines: IQRs of horizontal and vertical positions, and of horizontal and wind direction positions of calling birds at call onset, their flock mates at call onset and all birds during four flight sessions, respectively; the lines' intersections are at the medians of the distributions; thin black lines: flight section's outline.

flight makes them conspicuous to predators, whereas a high call rate while hidden in vegetation does not.

Comparable to the vocal behaviour of wild zebra finches[16], at take-off from the perch our birds often emitted a stack call, which could be followed by a distance call shortly after. During sustained flocking flight, mainly stack calls (Fig. 3b) were emitted. To test for flocking flight specificity of calling behaviour, we radio-telemetrically recorded the individual vocal activity during four solo flight sessions per bird. As in flocking flight, calling activity in solo flight was maximal during the first four seconds after take-off. In contrast to flocking flight, however, birds flying solo in the flight section never vocalized during the phase of sustained flight (Fig. 3a, bottom panel). This indicates that the emission of calls during sustained flight in zebra finch flocks depends on the social context. During both flight phases, take-off and sustained flight, zebra finches called significantly less when flying solo (population mean ± standard deviation (s.d.): 0.29 ± 0.23 Hz and 0 ± 0 Hz for take-off and sustained flight, respectively; $n = 19$ sessions) than when flying in a flock (population mean ± s.d.: 0.47 ± 0.39 Hz and 0.04 ± 0.06 Hz for take-off and sustained flight, respectively; $n = 65$ sessions; Extended Data Fig. 6). Although calls were most frequently emitted during the take-off phase of the flight session, we restricted

further analysis to stack calls emitted during the sustained phase of flight. The effect of take-off calls and of distance calls, which are very rarely emitted during the sustained phase of flocking flight, on flock organization still needs to be investigated.

In addition to the general social context, the spatial arrangement of birds in the flock might also affect a bird's propensity to emit a call during flight in the flight section. We compared the spatial positions of calling birds at stack call onset with the spatial positions of their flock mates, and found that, indeed, at call onset the calling bird was located at the right, lower edge of the frontal part of the flock (Fig. 3c,d). For example, bird Green, the individual that emitted the majority of stack calls (60 out of 93) during flocking flight, was also most often located at the bottom edge of the frontal part of the flock (Fig. 1d and Extended Data Fig. 1).

**Vocally guided vertical repositioning.** To determine if call emissions during sustained flight are correlated with a bird's flight behaviour, we tracked (sample rate: 24 Hz) the calling bird's spatial position relative to its position at call onset in the synchronized footage of both cameras. Following the onset of every stack call ($n = 93$) emitted during the phase of sustained flight in 12 flocking flight sessions, we performed the tracking in every 5th

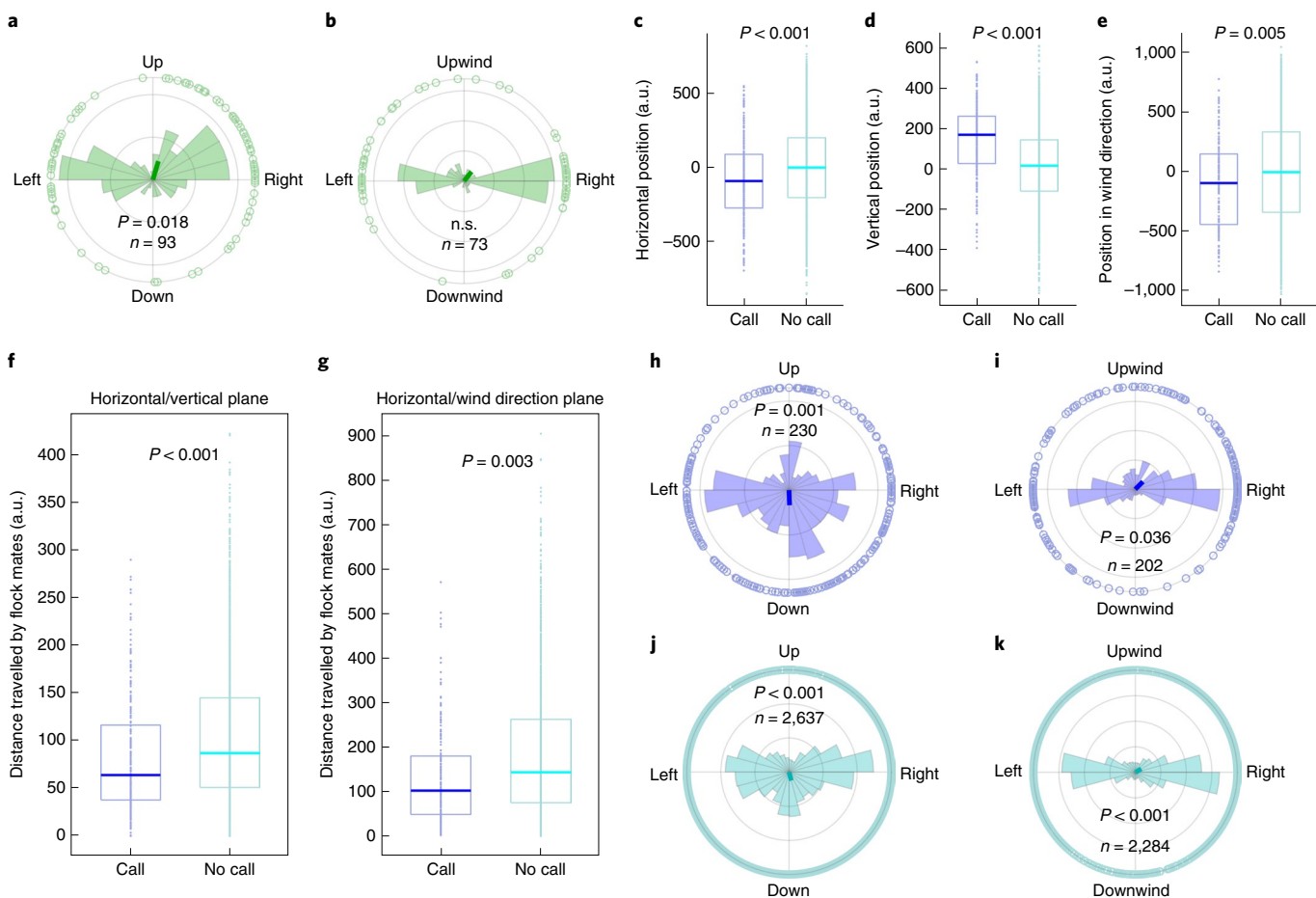

**Fig. 4 | Movement directions of calling birds and reactions of flock mates. a,b,** Movement directions of a calling bird within 209 ms following the onset of stack calls showed a significant (Rayleigh test, $P = 0.018$, $z = 3.99$, $n = 93$) directionality in the horizontal/vertical plane (**a**), but not (Rayleigh test, $P = 0.277$, $z = 1.29$, $n = 73$) in the horizontal/wind direction plane (**b**). Light green circles and fans: individual movement directions and counts, respectively; dark green line: median movement direction. **c–e,** Box plots of flock mates' spatial positions relative to a focal bird's spatial position at the onset of 46 stack calls (blue) and at the time of initiation of 579 call-unaccompanied upwards movements (cyan). Dots: individual data points; boxes: 25th and 75th percentiles of distributions; horizontal lines: medians. LMM, estimates ± s.e.m.: $80.5 \pm 19.9$, $P < 0.001$, $t = 4.04$ (**c**); $-127.2 \pm 12.5$, $P < 0.001$, $t = 10.17$ (**d**); $105.5 \pm 37.9$, $P = 0.005$, $t = 2.78$ (**e**). Sample sizes in **c** and **d** as in **f**, sample sizes in **e** as in **g**. **f,g,** Call-accompanied upwards movements caused flock mates to significantly reduce movement activity in the horizontal/vertical (LMM, estimates ± SE: $20.6 \pm 4.9$, $p < 0.001$, $t = 4.2$, $n = 230$; **f**) and the horizontal/ wind direction plane (LMM, estimates ± s.e.m.: $52.3 \pm 10.2$, $P < 0.001$, $t = 5.2$, $n = 202$; **g**). Box plots show distances travelled by flock mates within 209 ms following the initiation of 46 call-accompanied (blue) and 535 call-unaccompanied upwards movements (cyan). Dots: individual data points; boxes: 25th and 75th percentiles of distributions; horizontal lines: medians; $P$ values of LMMs are indicated. **h–k,** Movement directions of upwards-moving birds' flock mates within 209 ms after call onset (**h** and **i**; Rayleigh test, $P = 0.001$, $z = 7.13$, $n = 230$, and $P = 0.036$, $z = 3.31$, $n = 202$, respectively) differ from movement directions of upwards-moving birds' flock mates within 209 ms after initiation of call-unaccompanied upwards movements (**j** and **k**; Rayleigh test, $P < 0.001$, $z = 27.43$, $n = 2,637$, and $P < 0.001$, $z = 15.05$, $n = 2,284$, respectively). Light blue and light cyan circles and fans: individual movement directions and counts, respectively; dark blue and dark cyan lines: median movement direction.

of 25 consecutive frames, which covered a time period of 209 ms. Our analysis showed that in the calling bird, the onset of a stack call emission was followed by an upwards-directed vertical position change (Fig. 4a,b, Extended Data Fig. 7 and Supplementary Video 3). The distribution of movement directions of calling birds within 209 ms after call onset showed a significant directionality in the horizontal/vertical plane, with the mean direction of movement pointing 73.5° upwards (Fig. 4a), but not in the horizontal/wind direction plane (Fig. 4b). Interestingly, the number of upwards-directed position changes accompanied by a stack call was very low in comparison with the number of upwards-directed position changes (that is, position change of at least 24.88 arbitrary units (a.u.) in 209 ms) not accompanied by a call. On average, only 1 in 94 ($n = 4$ flight sessions) upwards-directed position changes was accompanied by a call, which opposes the

hypothesis that call emission is only a byproduct of the motor act of moving upwards.

The propensity to move upwards was generally high in a bird that was flying at low vertical positions in the flight section (Extended Data Fig. 8). However, the propensity to emit a stack call prior to an upwards-directed movement depended on the spatial arrangement of birds in the flock during flight. While at the time of initiation of call-unaccompanied upwards movements the moving bird was evenly surrounded by its flock mates in all three dimensions (Extended Data Fig. 8c,d), at the initiation of call-accompanied upwards movements flock mates were clustered above, to the left and in the back of the calling bird (Extended Data Fig. 8a,b and Supplementary Information). Relative to the spatial position of an upwards-moving bird at the time of movement initiation, the bird's flock mates were located significantly more left, higher and more

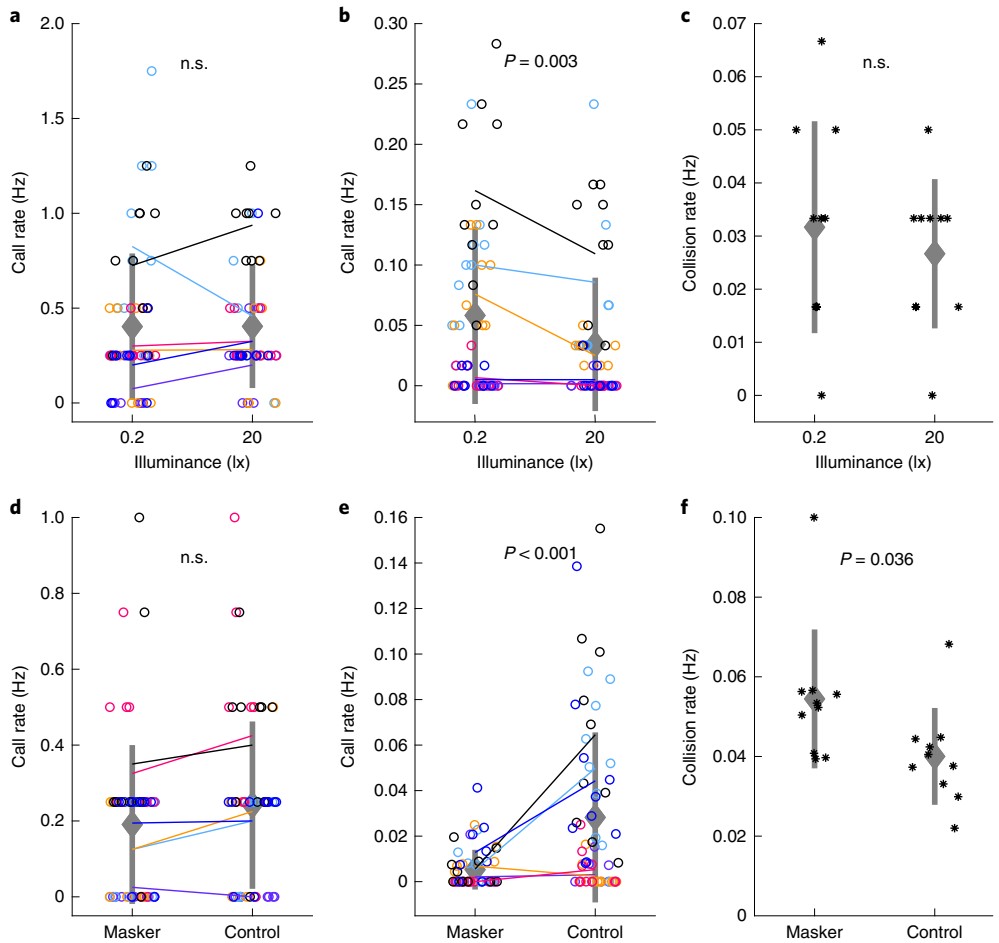

**Fig. 5 | Vision and vocal communication play a role in collision avoidance during flocking flight. a,b**, In contrast to the take-off phase (LMM, estimates ± s.e.m.: 0.03 ± 0.05, $P = 0.517$, $t = 0.65$; **a**), call emission rates during sustained flocking flight are affected by the ambient illuminance level (LMM, estimates ± s.e.m.: −0.02 ± 0.01, $P = 0.02$, $t = −2.33$; **b**). Coloured circles and lines indicate individual data points and means, respectively. Colours represent bird ID. Grey diamonds and thick lines mark population means ± s.d., respectively. $n = 60$ for each light condition. **c**, The rate of collisions between birds during flocking flight is not affected by the ambient illuminance level (LMM, estimates ± s.e.m.: −0.01 ± 0.01, $P = 0.5$, $t = −0.68$). Black asterisks mark individual data points. Meaning of remaining markers as in **a**. $n = 10$ per illuminance level. **d,e**, In contrast to the take-off phase (LMM, estimates ± s.e.m.: 0.05 ± 0.04, $P = 0.172$, $t = 1.37$; **d**), call emission rates during sustained flocking flight are affected by the presence of masking noise (LMM, estimates ± s.e.m.: 0.02 ± 0.004, $P < 0.001$, $t = 5.22$; **e**). Meaning of colours and markers as in **a**. $n = 60$ for each noise condition. **f**, The rate of collisions between birds during flocking flight is significantly affected by the presence of masking noise (LMM, estimates ± s.e.m.: −0.014 ± 0.006, $P = 0.036$, $t = −2.26$). Meaning of markers as in **c**. $n = 10$ per noise condition. At the top of each panel, the $P$ value of a linear mixed effects model is provided.

downwind when the bird was calling than when the bird moved upwards without calling (Fig. 4c–e). We therefore conclude that the propensity of a bird to emit a call during sustained flocking flight depends on the coincidence of two factors: (1) the bird is positioned at the bottom, right edge of the frontal part of the flock; and (2) the bird intends to move upwards.

The visual field of zebra finches is characterized by a blind area of about 60° located behind the head[22]. In addition, zebra finch vision seems to be impaired in a large part of the dorsal visual field by an intravitreous structure, the pecten oculi[23]. Furthermore, it has been shown that in some bird species, including zebra finches, vision is lateralized, with visual input from the right eye having higher behavioural relevance than visual input from the left eye[31,32]. Based on this knowledge, we assume that zebra finches flying at low and upwind positions at the right side of the flock are sometimes not able to visually localize all their flock mates. When a bird in such a situation intends to change its spatial position within the flock in an upwards direction, it emits a stack call to actively communicate its intention to its flock mates.

If vocal communication between the upwards-moving bird and its flock mates indeed occurs, and the flock mates perceive and evaluate the information carried by the stack calls, they should show a behavioural response temporally correlated to the calls. To test whether flock mates show a reaction to stack calls in their flight behaviour, we compared the flight behaviour of flock mates of birds moving upwards after calling with the flight behaviour of flock mates of birds that moved upwards without calling. We found that in contrast to flock mates of birds moving upwards without calling, call emission of an upwards-moving bird reduced movement activity in its flock mates (Fig. 4f,g). Within a time period of 209 ms after call onset and initiation of upwards movements, flock mates of calling and upwards-moving birds travelled over significantly shorter distances (population median and IQR: 64.1 and 37.9–116.6 a.u., and 104.1 and 50.6–181.8 a.u. for the horizontal/vertical plane and the horizontal/wind direction plane, respectively) than flock mates of birds that moved upwards without calling ($n = 2,637$ and $n = 2,284$, respectively; population median and IQR: 87.1 and 51.1–145.2 a.u., and 145.3 and 77–264.2 a.u. for the horizontal/vertical

plane and the horizontal/wind direction plane, respectively). This suggests that flock mates react to the calls by retaining their spatial position in the flight section, probably to observe the calling bird's movement and thus to reduce collision risk.

For both cases—call-accompanied and call-unaccompanied upwards movements—the flight trajectories of the moving bird's flock mates showed a significant directionality in both the horizontal/vertical plane and the horizontal/wind direction plane (Fig. 4h–k). Although in both conditions the main movement direction of an upwards-moving bird's flock mates was opposite to the upwards-moving bird, when the bird called before moving upwards a considerable amount (8.3%) of the calling bird's flock mates aligned their movements with those of the calling bird and moved towards positions between 75° and 105° upwards (Fig. 4h). Interestingly, these straight upwards-directed movements were particularly underrepresented in flock mates of birds moving upwards without calling (Fig. 4j). This suggests that in some of the upwards-moving bird's flock mates, the combination of an upwards movement with a stack call, but not the upwards movement alone, causes an upwards-directed movement and consequently an alignment of movement direction with the calling bird.

**Complementary role of vision and vocal communication.** To directly test for the effect of deficient visual input on the vocal activity and flight behaviour of zebra finches in flocking flight, we reduced the illumination in the flight section (initially 200 lx) and recorded the individual vocal behaviour of zebra finches during ten flocking flight sessions at low (20 lx, comparable to illumination levels during civil twilight) and during ten flocking flight sessions at very low (0.2 lx, comparable to the illumination of a clear night sky at full moon) illumination. While wild zebra finches may occasionally fly under 20 lx illumination levels, flying under very low illumination levels (0.2 lx), which in nature are present only during the night, is a very artificial situation for the diurnal zebra finch. Our experiment showed that while during the take-off phase the call rate was not affected by the illuminance level, during sustained flight birds called significantly more often when flying under very low light levels (population mean call rate ± s.d.: $0.06 \pm 0.07$ Hz; $n = 60$) than when flying under low light levels (population mean call rate ± s.d.: $0.03 \pm 0.06$ Hz; $n = 60$; Fig. 5a,b). Interestingly, the rate of collisions between birds did not differ between illuminance levels (Fig. 5c). The elevated call emission rates at very low ambient light levels in combination with the constant collision rates suggest that zebra finches may be able to compensate for deficient visual information by increasing the usage of vocal communication to coordinate their spatial positions in flying flocks.

Finally, to investigate the importance of vocal communication for flock coordination, we manipulated the birds' ability to communicate vocally during flight. In zebra finches, background noise can affect the call detection and discrimination ability, especially when the noise energy is within the calls' spectral region[33]. Stack and distance calls of zebra finches have their main energy roughly between 2,000 and 5,000 Hz (ref. [16]). Via a loudspeaker positioned on the floor of the flight section, we introduced additional noise to the flight section during flight sessions (Supplementary Information). We measured call emission and collision rates during ten flocking flight sessions carried out in the presence of band-pass filtered (cut-off frequencies: 1,500 and 8,000 Hz) additional noise that completely masked the birds' calls (Extended Data Fig. 9) and compared them with call emission and collision rates measured during ten flocking flight sessions carried out in the presence of low-pass filtered (cut-off frequency: 2,500 Hz) additional noise that did not mask the birds' calls. While the additional noise did not affect the call rate during the take-off phase, during the phase of sustained flight birds called significantly less when flying in the presence of masking noise (population mean call rate ± s.d.: $0.004 \pm 0.01$ Hz; $n = 60$) than when flying

in the presence of noise that did not mask their calls (population mean call rate ± s.d.: $0.024 \pm 0.038$ Hz; $n = 60$; Fig. 5d,e). Collision rates (see Supplementary Video 5 for an exemplary collision event) were significantly higher during flight in the presence of masking noise (mean collision rate ± s.d.: $0.054 \pm 0.017$ Hz; $n = 10$) than during flight in the presence of noise not masking the calls (mean collision rate ± s.d.: $0.04 \pm 0.012$ Hz; $n = 10$; Fig. 5f).

We conclude that not being able to vocally communicate with each other has a strong effect on the coordination of spatial positions of individuals in a zebra finch flock during flight. Deficits in social information influx resulting from not being able to see each other, however, can be compensated for by increasing the usage of vocal communication to reduce collision risk in the flying flock.

## Discussion

The results presented here are based on experiments carried out with a small flock of zebra finches flying under artificial conditions in the flight section of a wind tunnel. Whether our findings are transferrable to other species or larger bird flocks flying in their spatially unrestricted natural environment has yet to be proved. However, investigating how organisms exchange information that enables them to precisely coordinate their spatial positions during collective movements is technically challenging[34]: the quantity and quality of social information available to each individual has to be continuously measured with high spatial and temporal resolution for different sensory modalities in several individuals synchronously, and in parallel to each individual's spatial position within the moving group. Acquiring such data over extended time periods in animal groups that move freely in open space requires the recording equipment to move with the moving group. Although recent technological advancements opened up the possibility to track collective behaviour at large spatiotemporal scales in field-based settings[20,35], to the best of our knowledge, there is no technology available yet that would render possible the long-term synchronous recording of head orientation, vocal activity and spatial position of each individual in a flock of songbirds flying freely under the open sky.

By uncovering the sensory modalities zebra finches exploit to coordinate their spatial positions in a flying flock, our study advances the current knowledge on the mechanisms that underlie collective behaviour. Most interestingly, we can demonstrate a vital role of vocal communication for the spatial coordination of birds in moving groups. This finding indicates that in contrast to common assumption, individuals in moving groups of birds not only watch the behaviour of other group members but can actively exchange directional information to enable adequate group performance. Here, we report on local interactions between individual birds. Whether vocal communication can be beneficial also for the spatial coordination of individuals during large-scale collective behaviour remains to be determined. Nevertheless, our study encourages a revision of current theoretical models on animal collective behaviour and an integration of vocal communication as a potential mechanism for group coordination.

## Methods

**Birds and ethical approval.** All zebra finches (*Taeniopygia guttata*) used for data collection originated from a breeding colony at the Max Planck Institute for Ornithology in Seewiesen, Germany. The birds were housed together in a large (2 m high × 1 m wide × 2 m long) indoor aviary in the vicinity of the wind tunnel's flight section. This setup enabled the birds to enter the flight section by themselves without additional handling by the experimenter. Illumination of the aviary followed a 12 h light/dark cycle and all birds had ad libitum access to food (seed mix, greens, boiled eggs) and water. After each flight session, the birds received seed heads of foxtail millet (*Setaria italica*) as an additional food reward. Birds were colour banded for individual recognition.

All experiments were performed in accordance with the current version of the European law on animal protection, and experimental protocols have received ethics approval by the Government of the Free State of Bavaria, Germany (reference number: 55.2-2532.Vet_02-17-211).

**Flight sessions.** For data collection, a group consisting of one adult male and five adult female zebra finches flew together for one to two minutes in a low-speed subsonic wind tunnel (SWT112R Race Track) located at the Max Planck Institute for Ornithology in Seewiesen, Germany. Wind speed and temperature in the flight section (1.1 m high × 1.2 m wide × 2.0 m long) were automatically adjusted to 10 m s⁻¹ and 18 °C, respectively. To facilitate the development of flight muscles and physiological capacity, birds were first trained individually to fly continuously for several minutes in the flight section. Each bird autonomously entered the flight section and sat down on a perch to initiate a flight session. The wind speed was slowly increased to 10 m s⁻¹, a speed at which the body lift is maximal in zebra finches flying in a wind tunnel[36]. After the bird became airborne, the perch was removed from the flight section. To prevent birds from landing on the flight section's floor during the flight session, a landed bird was gently nudged at the wings or the tail with a padded wooden stick. After three to four minutes or when the bird attempted to land frequently, the perch was reintroduced into the flight section to signal the end of the flight session. When the bird had again landed on the perch, the wind was switched off and the bird was allowed to fly back to its aviary. Flight sessions were repeated two to three times per day with intermittent resting periods of at least two hours.

When the training phase was completed, the birds were allowed to enter the flight section as a flock, and data acquisition started. We recorded each bird's individual vocal behaviour simultaneously with its head orientation and 3D spatial position during 13 flocking flight sessions. After finishing data acquisition for the flocking flight condition, we reran the experiment and interspersed flocking flight sessions with flight sessions in which each bird flew solo in the wind tunnel's flight section. Each bird's vocal behaviour was recorded simultaneously with its head orientation and 3D spatial position during four solo flight sessions per bird.

To test for the influence of ambient illumination levels on the birds' vocal behaviour and flight performance during flocking flight, illumination in the flight section (initially 200 lx) measured with a digital lux meter (Unitest 93560, Beha-Amprobe, Fluke) was reduced to 0.2 lx. During a five-day training phase, the birds flew together in the flight section under illumination levels that were gradually lowered each day. Subsequently, each bird's individual vocal activity and 2D spatial position was recorded during ten flocking flight sessions at 0.2 lx illumination, followed by ten flocking flight sessions at 20 lx illumination.

To investigate the importance of vocal communication for flock coordination, we carried out flocking flight sessions during which we masked the birds' calls by introducing additional noise to the flight section via a loudspeaker (iLoud, IK Multimedia) located on the flight section's floor below the flying flock. The frequency bandwidth of the masking noise was band-pass filtered between 1.5 and 8 kHz, which roughly corresponds to the frequency range of the zebra finches' calls. Presentation level was adjusted so that in this frequency range, the noise amplitude in the flight section at a wind speed of 10 m s⁻¹ was on average 20 dB higher than the amplitude of zebra finch calls emitted from stationary birds sitting on the perch in the calm flight section (Extended Data Fig. 9). As a control condition, flocking flight sessions were carried out while presenting low-pass filtered noise (cut-off frequency: 2.5 kHz), which did not additionally mask the zebra finches' calls. After a five-day training period with both types of noise, the birds were comfortable to fly in the flight section during the noise presentation. Subsequently, the birds' individual vocal activity and spatial positions in 3D were recorded during ten flight sessions for each noise condition. Noise conditions alternated during the data acquisition period in such a way that after an initial session with masking noise, two sessions with low-pass filtered noise always followed two sessions with masking noise.

It is important to note that flock composition differed slightly between the data acquisition periods. The male bird Green died due to a gut tumour a few months after data acquisition for the first and second dataset (flocking flight at 200 lx and solo flight) had been finished. The bird was replaced by the male bird Dark blue for the periods of data collection for the third (flocking flight at 20 and 0.2 lx) and fourth (flocking flight in the presence of additional noise) dataset. Owing to the change in flock composition, data from acquisition periods one and two are not directly compared with data from acquisition periods three and four here.

**Video recording.** Spatial positions and head orientation of zebra finches during flight in the wind tunnel's flight section at normal (200 lx) illumination was filmed using two digital video cameras. Camera 1 (HERO5 Black, GoPro) was fixed in the centre of the flight section's downwind end to record the birds' spatial positions along the flight section's horizontal and vertical dimension (Fig. 1a). This camera was set to a video resolution of 1,080 px at 120 frames per second, a screen resolution of 1,920 × 1,080 px at an aspect ratio of 4:3, and a narrow field of view. Camera 2 (HERO8 Black, GoPro) was placed at the floor of the flight section below the flying birds to record their positions in horizontal and wind direction and their head orientation (Fig. 1a). Except for the field of view, which was set to wide in Camera 2, settings were equal to Camera 1. While birds were constantly in the field of view of Camera 1 throughout all sessions, single birds were able to transiently leave the field of view of Camera 2 when flying. On average, a bird's position in wind direction could be determined for 77.1% of the flight time. Video recording was started in both cameras via remote control when all birds had sat down on the perch in the flight section. For each flight session, a video file in MP4 format was

saved on a microSD card in each camera. The audio tracks of the MP4 files were used to record the overall soundscape in the flight section via the cameras' built-in microphones (stereo, sample rate: 44.1 kHz).

For video recording (29.97 frames per second, 1,280 × 720 px) the birds' flight behaviour under low light (0.2 and 20 lx) conditions, the GoPro Camera 1 was replaced by a digital night vision camera (Aurora Black, SIONYX), and the birds' spatial positions were only recorded in 2D.

**Audio recording.** To record the birds' individual vocal activity while flying in the wind tunnel's flight section, at the start of each data acquisition period each bird was equipped with a light-weight radio-telemetric microphone transmitter, developed and manufactured at the Max Planck Institute for Ornithology in Seewiesen, Germany[27]. The transmitter (weight: 0.6 g), which included a miniature condenser microphone (FG-23329, Knowles Electronics) and a battery (Zinc Air P10, Duracell), was covered by a thin silicon casing and was fixed on the bird's back with cotton-covered rubber band loops around both femurs and the abdomen. As the microphone's sensitive side was directed towards the bird's body, the transmitter and overlaying feathers shadowed the microphone from wind noise and sounds generated by nearby conspecifics. Microphone transmitters remained on the bird until the end of the data acquisition period. For the detection of the transmitted signals, a crossed Yagi antenna (Winkler Antennenbau) was placed on top of the flight section. An antenna amplifier (TVS 14-00 Axing, Goobay) increased the antenna signal by 18 dB. The incoming signal was split (BE 2-01 premium-line) and fed into six communication receivers (AOR 8600, AOR), which were modified to handle 12 kHz audio bandwidth. The analogue signals were digitized by an eight-channel external soundcard (sampling rate: 22.05 kHz; M-Track Eight, M-Audio), which was connected to a laptop computer (Latitude E5450, Dell Technologies). During flight sessions, all digitized microphone signals were recorded in parallel as continuous sound files in WAV format using multichannel software (16 bit, 22.05 kHz, ASIO, Steinberg Media Technologies). This procedure allowed us to unambiguously assign each single vocalization to the one bird it was emitted from while preserving the precise temporal relationship between vocalizations of all birds. Audio recording was started right before the birds entered the flight section.

At the beginning and at the end of each flight session, while the birds were sitting on the perch in the flight section, each bird's ID was first pronounced loudly by the experimenter in the order the birds were placed on the perch, followed by a loud clapping of the experimenter's hands. This 'vocal clapperboard' had two functions: (1) to recognize individual birds in the video files; and (2) to synchronize the sound files with the audio tracks of the video files during data analysis.

**Data analysis.** An overview of all numbers of recorded and analysed flight sessions can be found in Supplementary Table 2.

To synchronize both video files, for each flight session a multicam clip was generated from both video files with the video editing software Final Cut Pro X (v. 10.4.8, Apple). With the help of the vocal clapperboard, the video files were temporally aligned and subsequently synchronized with the six audio files.

To describe the general flight behaviour of zebra finches during flocking and solo flight in the flight section of the wind tunnel, the spatial position of each bird was tracked (Tracker, v. 5.1.4, Open Source Physics, https://physlets.org/tracker; sample rate: 24 Hz) in both synchronized video files throughout the entire duration of four flight sessions per experimental condition (flocking flight sessions 2, 5, 8 and 13, and solo flight sessions 1–4). The four flocking flight sessions were chosen so that analysed sessions were roughly equally distributed over the time period of data acquisition. Flight paths for each bird and each flight session were reconstructed in three dimensions, with coordinates in the horizontal and vertical dimension provided by the footage taken with Camera 1, and with coordinates in the wind direction provided by the footage taken with Camera 2. Note that because the tracking software provided spatial positions in the form of pixel coordinates that are prone to parallax and perspective effects in the footage, we used arbitrary units for labelling in all figures displaying spatial data.

To determine the frequency of position change for each bird in completely tracked flight sessions, autocovariance sequences were computed (xcov function, MATLAB, MathWorks) from the reconstructed flight paths. Subsequently, a power spectral density estimate was calculated (periodogram function, MATLAB) to detect any periodicity in the autocovariance sequence, and the peak frequency in the power spectral density estimate was taken as the frequency of position change. Only peak values that exceeded five standard deviations of the mean of the whole spectrum were considered to be significant. Because birds were able to exit the field of view of Camera 2 and therefore data on a bird's position in wind direction could be noncontinuous throughout a flight session, the rhythmicity of positional changes in wind direction could not be determined.

To detect time periods in flocking flight sessions in which movements were aligned between birds, a correlational analysis of movement directions (comparable to the analysis described in ref. [18]) was performed with a moving time window. For each time step (sample rate: 24 Hz) in each flocking flight session and for each pairwise combination of birds, the circular correlation coefficient of flight directions within a time window of 498 ms was calculated (circ_corrcc function, Circular Statistics Toolbox, MATLAB[37]).

To measure the direction of head turns relative to the wind direction during horizontal position changes, for both flocking flight and solo flight, and each bird, ten video sequences were chosen in which the bird either moved to its right or to its left within the flight section. We randomly chose one flight session per condition and analysed for each bird the first ten horizontal position changes in the footage taken with Camera 2, in which all three points (the bird's beak tip, neck and base of the tail) were clearly discriminable from the background. For each horizontal position change, the spatial position of the bird's beak tip, neck and base of the tail was tracked (Tracker, sample rate: 120 Hz) from the start until the end of the position change. The start of a horizontal position change was defined as the first time point when the bird either turned its head more than 10° or changed its horizontal position more than 5%. The end of a horizontal position change was defined as the time point when the bird did not move more than 5% from its final destination (see Fig. 2e for an example). To determine the median angle of head turn relative to the wind direction in the flight section, the median difference between the bird's beak tip and neck positions in horizontal and wind direction was calculated over all sampled time points during the horizontal position change and transferred into an angular measure. To determine the direction of position change, relative to the wind direction in the flight section, the difference between the bird's neck positions in horizontal and wind direction at position change on- and offset was calculated and transferred into an angular measure.

The time points of call emission onsets relative to take-off were audio-visually determined from a spectrogram (Hamming window with a size of 512 samples) generated for each bird's sound file of each flight session with the free multi-track audio editing software Audacity (v. 2.4.2, Audacity Team, https://audacityteam.org/). As the vocal recordings of bird Black for most solo and flocking flight sessions were too noisy to detect vocalizations, vocal data from this bird were excluded from further analysis. For the same reason, one solo flight session of bird Green was also excluded from sound analysis. Call types were assigned to detected calls according to spectral and temporal characteristics of zebra finch call types described in ref. [15].

To test for a correlation between stack calling and flight behaviour of birds, vocalization-triggered flight paths were measured in three dimensions. For each stack call ($n = 93$) detected in a bird's microphone signal throughout 12 flight sessions, the spatial position of the calling bird and of all other flock members was manually tracked (sample rate: 24 Hz, Tracker) in the synchronized video files of both cameras for a time period of 209 ms after call onset. Calls emitted during flight session 13 were excluded from this part of the analysis, because the individual identification in the video footage was not possible for all birds throughout this session.

In the video files of each flocking flight session, the number of collisions between two or more birds was counted by visually examining the video material in slow motion with Tracker. Each event in which two or more birds touched each other during flight, causing a change in flight direction in at least one of them, was counted as a collision (see Supplementary Video 5 for an example).

**Statistics.** All statistical analyses were done with the Statistics Toolbox for MATLAB and the Circular Statistics Toolbox for MATLAB[35]. Training zebra finches to fly in the wind tunnel is time consuming and not every bird is able to perform the task. Therefore, we chose a repeated measure design for our study, and collected data from only one small flock of six zebra finches. To test for significance of observed effects at the population level, linear mixed models (LMMs) were fitted to the data (fitlme, MATLAB), which accounted for non-independencies due to repetitions of measurements taken from the same animals on consecutive days. In all models, the different flight sessions and individuals have therefore been considered as random effects.

**Reporting summary.** Further information on research design is available in the Nature Research Reporting Summary linked to this article.

## Data availability

Source data for all figures and Extended Data figures are provided with the paper. The raw data generated during this study are available from the corresponding author upon reasonable request.

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

## Acknowledgements

We thank H. Tsivlin for help with animal training, L. Trost for managing ethics approval, N. Heinecke, H. Sagunsky and M. Abels for technical support, P. D'Amelio for advice on statistics and U. Firzlaff for helpful comments on an earlier version of this paper. We acknowledge funding of the present work provided by the Max Planck Society to M.G.

## Author contributions

S.H. and C.R. conceptualized the project. M.G. was responsible for funding acquisition and provided resources. C.R. undertook project administration. S.H. supervised the project. Methodology was carried out by F.A., M.S.S., S.H. and C.R. Investigation was carried out by F.A., M.S.S., L.P., C.R. and S.H. F.A., M.S.S., L.P. and S.H. were responsible for data curation. Formal analysis was carried out by S.H. Visualization was carried out by S.H. and F.A. S.H. wrote the original draft of the manuscript. All authors reviewed and edited the final draft.

## Funding

## Competing interests

The authors declare no competing interests.

## Additional information

**Extended data** is available for this paper at https://doi.org/10.1038/s41559-022-01800-4.

**Correspondence and requests for materials** should be addressed to Susanne Hoffmann.

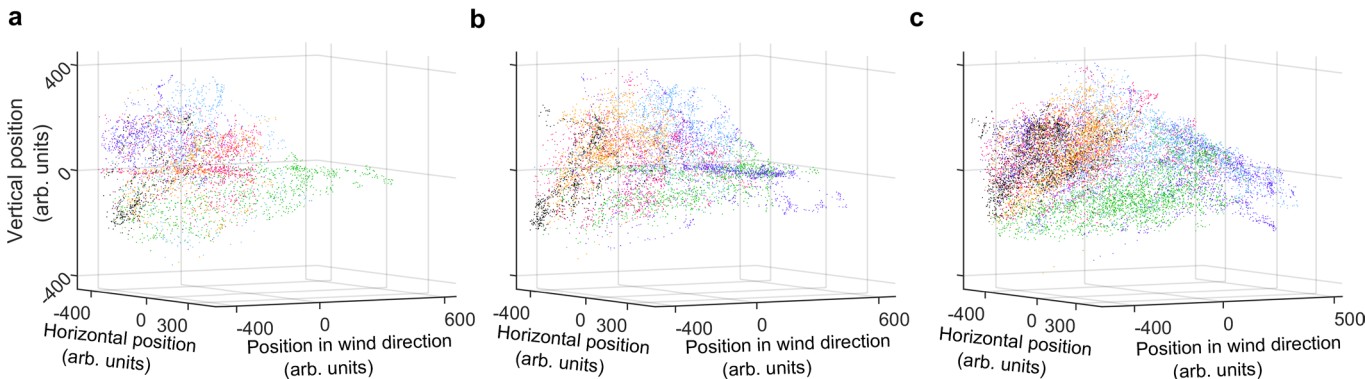

**Extended Data Fig. 1 | Positions of individual zebra finches in the flying flock. a–c**, Pseudo 3D representations of spatial positions (tracked with a sample rate of 24 Hz) of all birds within the flight section during flight session #2, #5 and #13, respectively. Marker color represents bird ID. Negative values correspond to bottom, left, and downwind positions in the flight section.

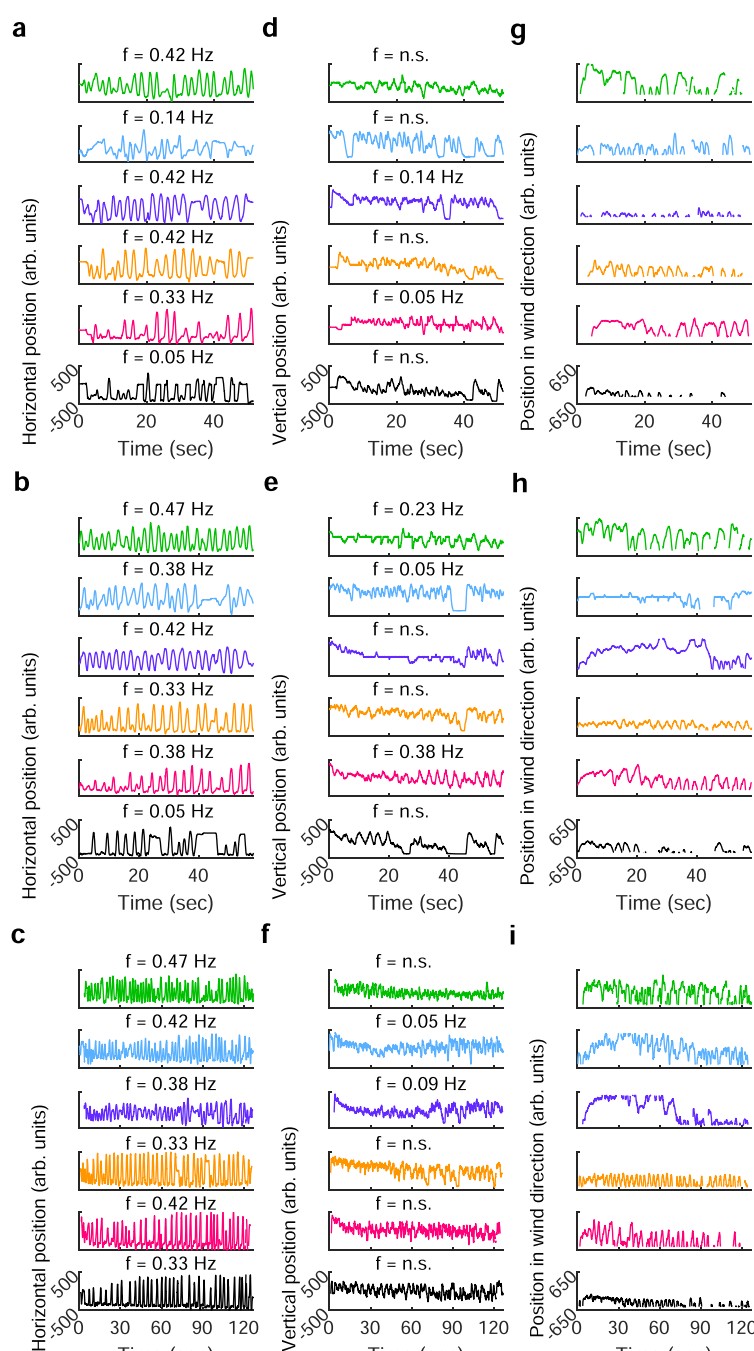

**Extended Data Fig. 2 | Flight paths of individual birds during flocking flight. a–c**, **d–f**, and **g–i**, Reconstructed flight paths (sample rate: 24 Hz) in horizontal, vertical and wind direction, respectively, of each bird during flocking flight session #2, #5 and #13 (top to bottom). f: frequency of direction change (cycles per second). n.s.: not significant. Line color represents bird ID. Negative values correspond to bottom, left, and downwind positions in the flight section.

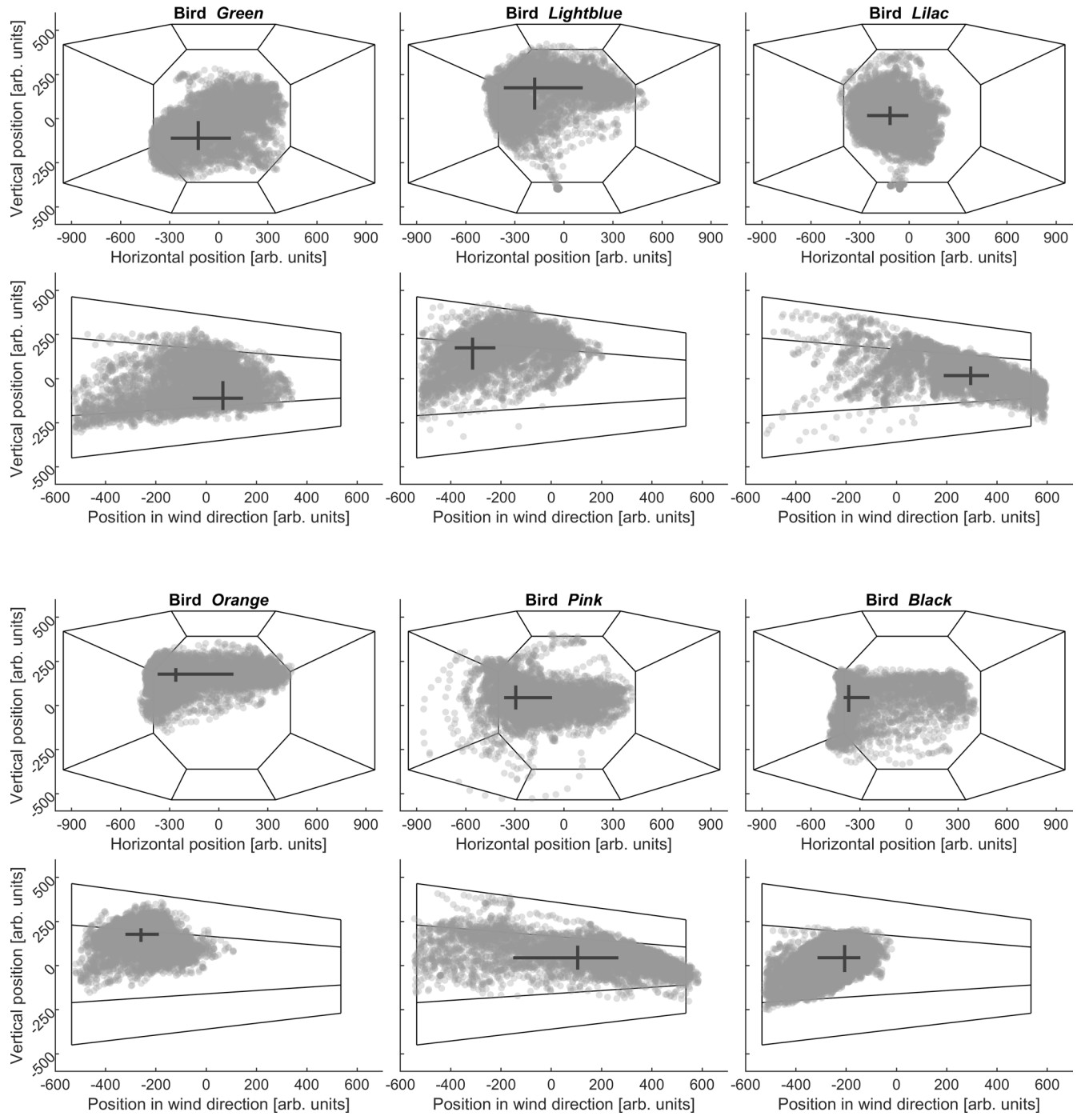

**Extended Data Fig. 3 | Positions of birds in the flight section during solo flight.** Light grey dots mark spatial positions at which a bird was detected during all four solo flight sessions. Dark grey lines indicate the interquartile range of horizontal, vertical and wind direction positions. The lines' intersections are at the medians of the distributions. Thin black lines indicate the flight section's outline.

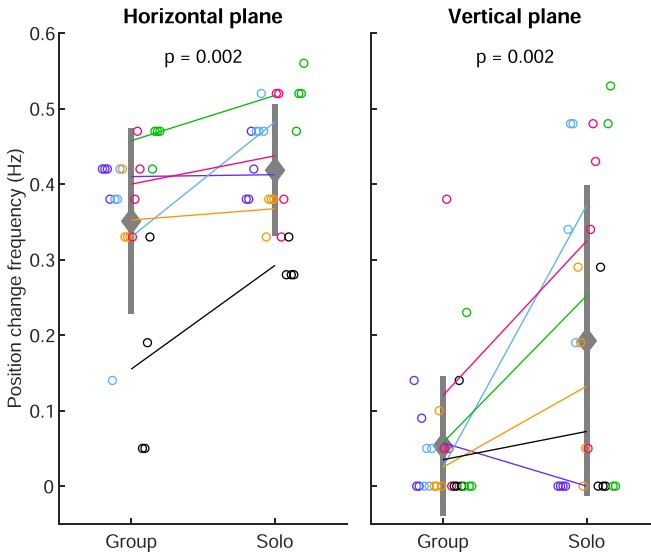

**Extended Data Fig. 4 | Frequencies of position changes in flocking and solo flight.** Colored circles and lines represent individual data points and individual means, respectively. Colors indicate bird ID. Grey diamonds and thick grey lines mark population means and standard deviation, respectively. Please note that the frequency of position change assumed the value zero in cases where no significant peak in the periodogram could be detected. Horizontal plane: LMM, estimates ± SE: 0.07 ± 0.02, $p = 0.002$, $t = 3.33$. Vertical plane: LMM, estimates ± SE: 0.14 ± 0.04, $p = 0.002$, $t = 3.27$. $n = 4$ sessions per bird and condition.

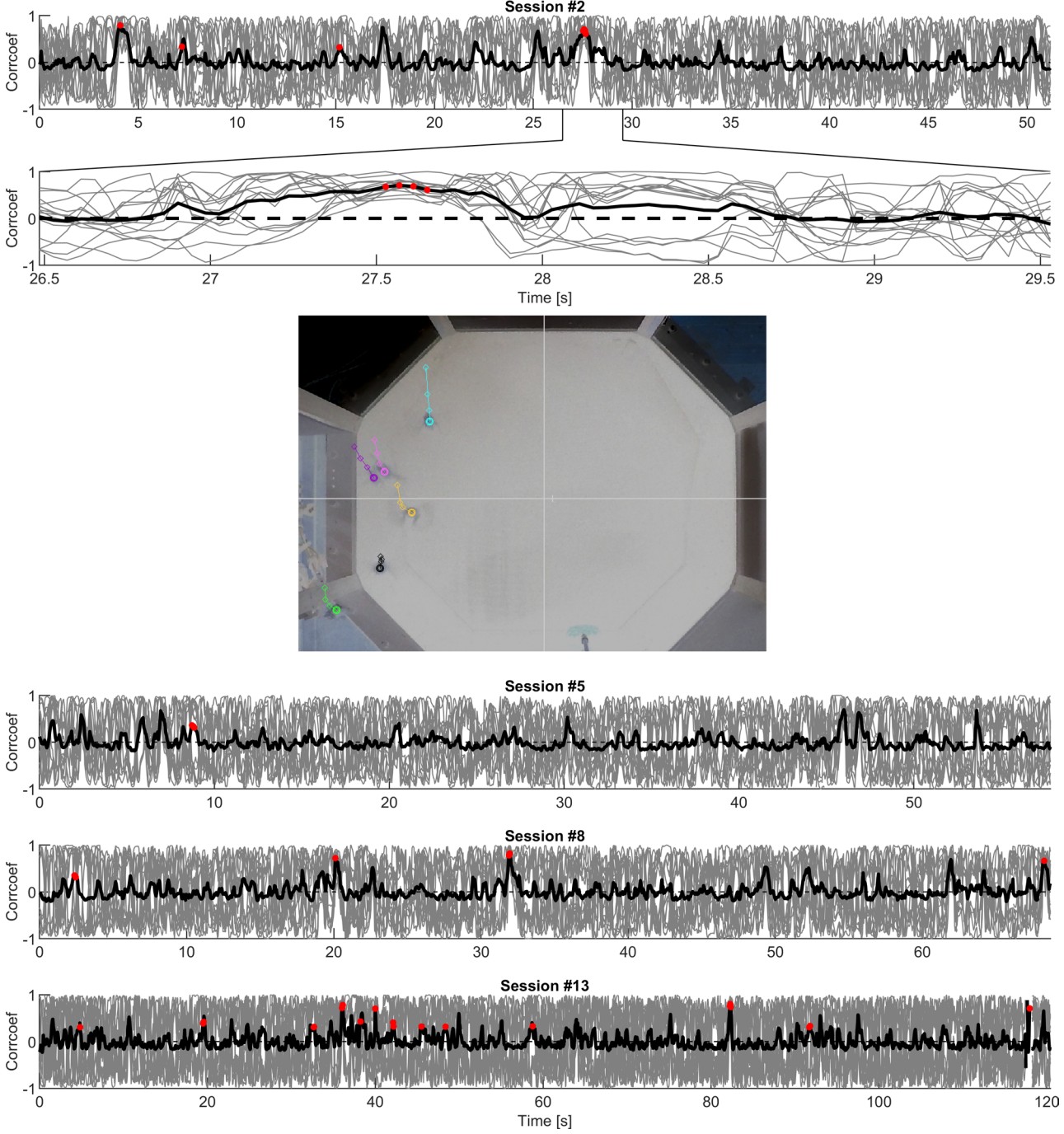

**Extended Data Fig. 5 | Alignment of movement directions between birds during flocking flight.** For four flocking flight sessions (session #2, #5, #8, and #13), correlation coefficients between movement directions of two birds in a pair are represented by grey lines for each of the 15 possible combination of birds. The mean correlation coefficient is indicated by the thick black solid line. The thin dashed black line marks the zero line. Red dots indicate time points at which movement directions were significantly aligned between the majority of birds (that is the mean correlation coefficient was larger than 0.3 and the mean p value was smaller than 0.05). In the freeze frame of the footage taken with Camera 1, the spatial positions corresponding to the time points marked by red dots in the panel above are shown for all flock members. Colored circles indicate the birds' current positions and the colored diamonds indicate the birds' positions at the three preceding time steps. Note the alignment of movement directions between all birds within this time period. Colors indicate bird ID. Wind direction is perpendicular to the plane of the image.

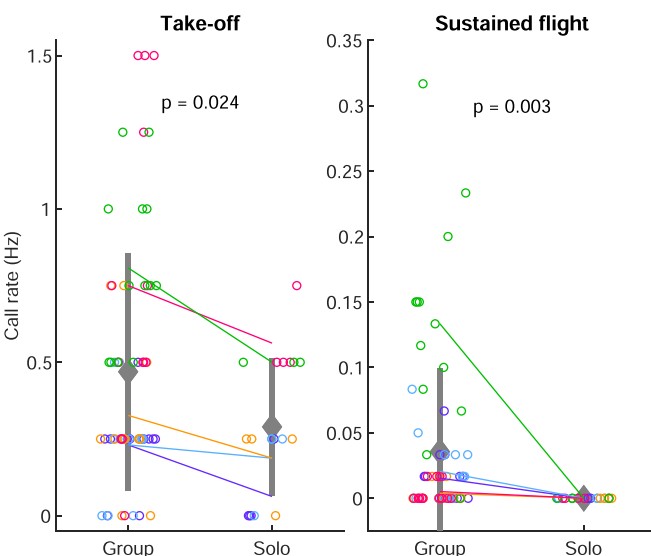

**Extended Data Fig. 6 | Call rates differ between flocking and solo flight.** Colored circles and lines represent individual data points and individual means, respectively. Colors indicate bird ID. Grey diamonds and thick grey lines mark population means and standard deviation, respectively. Take-off: LMM estimates ± SE: −0.16 ± 0.07, $p = 0.024$, $t = −2.29$. Sustained flight: LMM estimates ± SE: −0.04 ± 0.01, $p = 0.003$, $t = −3.1$. $n = 13$ sessions per bird for group flight and $n = 4$ sessions per bird for solo flight.

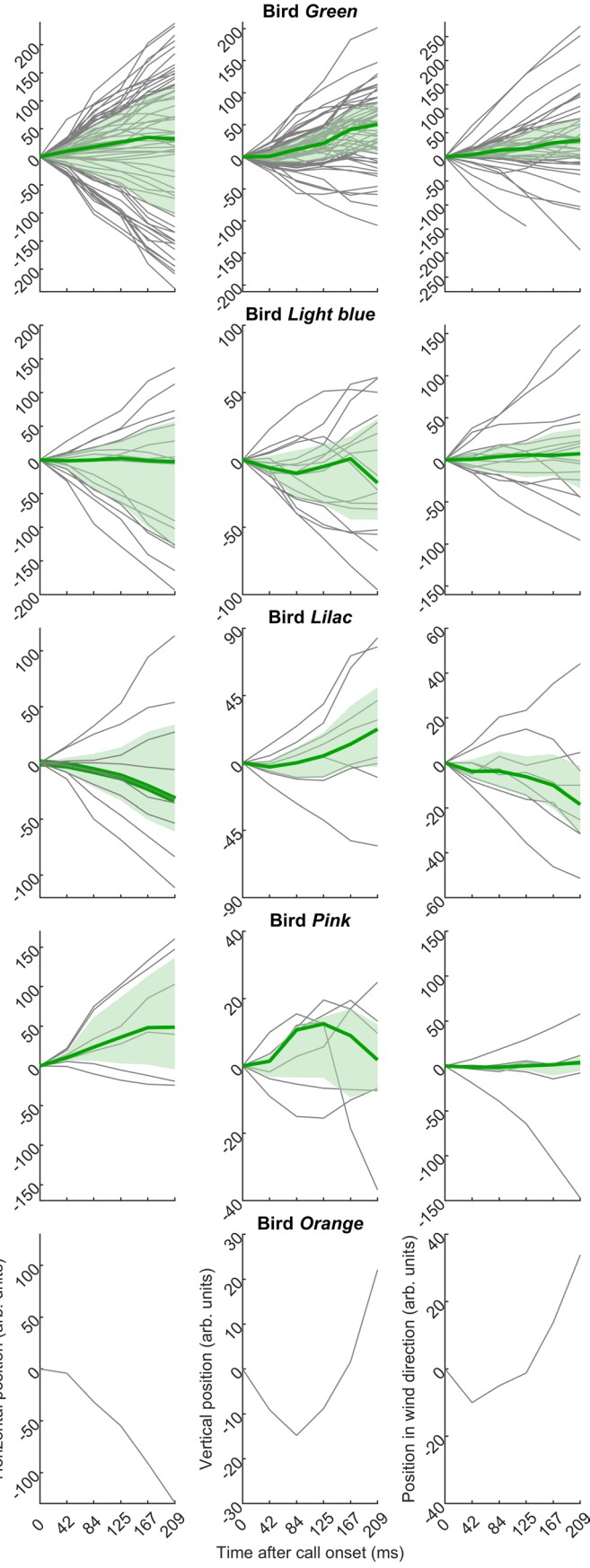

**Extended Data Fig. 7 | Flight paths of calling birds after call onset.** Spatial positions of calling birds relative to their spatial positions at call onset are shown in light grey for each individual, for all three dimensions, and for a time period of 209 ms (step widths: 41.5 ms) following the onset of Stack call emissions (Green: $n=60$, Light blue: $n=16$, Lilac: $n=9$, Pink: $n=7$, Orange: $n=1$). Dark green line: median; light green area: interquartile range.

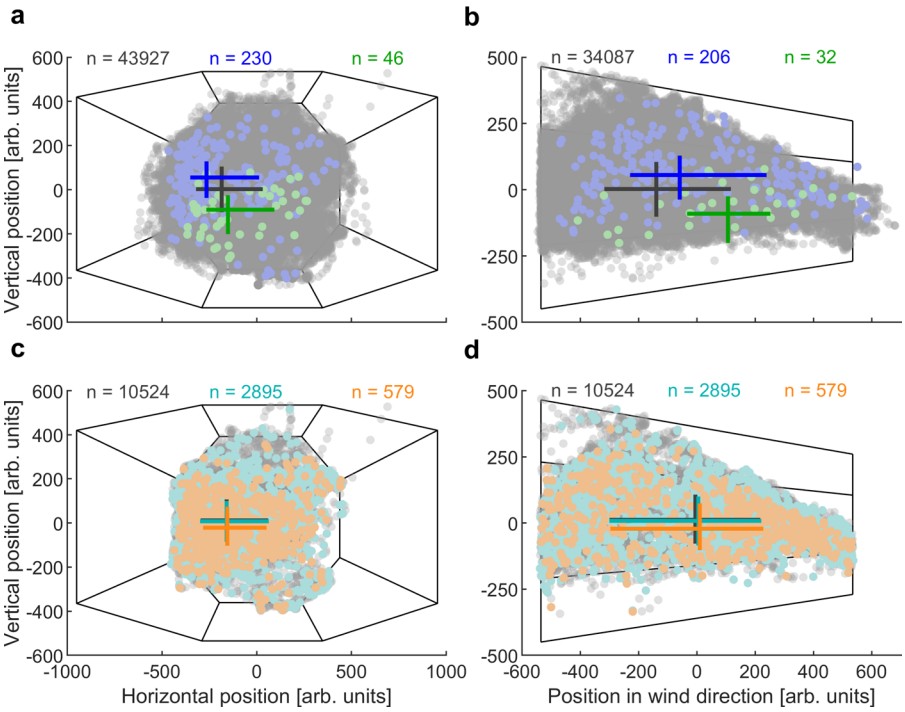

**Extended Data Fig. 8 | Spatial positions of birds at movement onset differ between call-accompanied and call-unaccompanied upwards movements.**
**a,b**, Spatial positions of calling birds (light green) and their flock mates (light blue) at the onset of call-accompanied upwards movements, and spatial positions of all birds at all time steps of four entire flocking flight sessions (light grey). Dark blue, dark green and dark grey lines indicate the interquartile ranges of horizontal and vertical positions, and of horizontal and wind direction positions of calling birds at call onset, their flock mates at call onset and all birds during four flocking flight sessions, respectively. The lines' intersections are at the medians of the distributions. **c,d**, Spatial positions of upwards moving birds (light orange) and their flock mates (light cyan) at the onset of call-unaccompanied upwards movements, and spatial positions of all birds at all time steps of four entire flocking flight sessions (light grey). Dark orange, dark cyan and dark grey lines indicate the interquartile ranges of horizontal and vertical positions, and of horizontal and wind direction positions of moving birds at movement onset, their flock mates at movement onset and all birds during four flight sessions, respectively. The lines' intersections are at the medians of the distributions. Thin black lines represent the flight section's outline. At the onset of both call accompanied (**a** and **b**; $n = 46$, 13 flight sessions) and call unaccompanied (**c** and **d**; $n = 579$, one flight session) upwards movements, the bird that moved upwards was located at significantly (LMM, estimates ± SE: 103.4 ± 22.3, $p < 0.001$, $t = 4.65$, for call-accompanied movements, and estimates ± SE: 13.0 ± 6.3, $p = 0.04$, $t = 2.05$ for call-unaccompanied movements) lower spatial positions than any bird was located throughout the flight sessions ($n = 43{,}927$ (four sessions) and $n = 10{,}524$ (one session) for call-accompanied and call-unaccompanied movements, respectively.

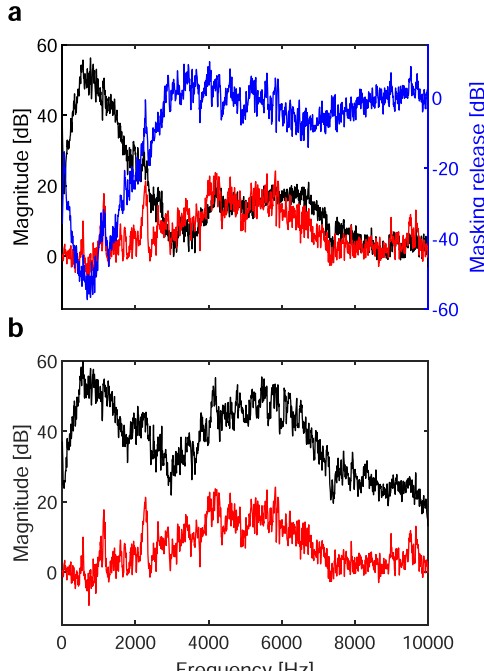

**Extended Data Fig. 9 | Call amplitudes exceed wind noise levels in the flight section but are completely masked by additionally presented band-pass filtered noise. a**, The mean magnitude spectrum of 18 150-ms long recordings from nine spatial positions in the flight section at a wind speed of 10 m/s, and of 18 Stack calls recorded 50 cm in front of the beak of zebra finches sitting on a perch in the calm flight section, is shown in black and red, respectively. The blue curve shows the difference of the two spectra. Please note that frequencies between three and six kilohertz in the spectrum of the Stack calls exceed the spectrum of the wind noise by up to 10 dB. **b**, When additional band-pass filtered noise was presented, calls were completely masked by the background noise in the flight section. The magnitude of the noise spectrum (black) exceeds the magnitude of the call's spectrum (red) at all frequencies.

# Reporting Summary

## Statistics

For all statistical analyses, confirm that the following items are present in the figure legend, table legend, main text, or Methods section.

| n/a | Confirmed | |
|---|---|---|
| ☐ | ☒ | The exact sample size (*n*) for each experimental group/condition, given as a discrete number and unit of measurement |
| ☐ | ☒ | A statement on whether measurements were taken from distinct samples or whether the same sample was measured repeatedly |
| ☐ | ☒ | The statistical test(s) used AND whether they are one- or two-sided *Only common tests should be described solely by name; describe more complex techniques in the Methods section.* |
| ☒ | ☐ | A description of all covariates tested |
| ☒ | ☐ | A description of any assumptions or corrections, such as tests of normality and adjustment for multiple comparisons |
| ☐ | ☒ | A full description of the statistical parameters including central tendency (e.g. means) or other basic estimates (e.g. regression coefficient) AND variation (e.g. standard deviation) or associated estimates of uncertainty (e.g. confidence intervals) |
| ☐ | ☒ | For null hypothesis testing, the test statistic (e.g. $F$, $t$, $r$) with confidence intervals, effect sizes, degrees of freedom and $P$ value noted *Give P values as exact values whenever suitable.* |
| ☒ | ☐ | For Bayesian analysis, information on the choice of priors and Markov chain Monte Carlo settings |
| ☒ | ☐ | For hierarchical and complex designs, identification of the appropriate level for tests and full reporting of outcomes |
| ☐ | ☒ | Estimates of effect sizes (e.g. Cohen's *d*, Pearson's *r*), indicating how they were calculated |

*Our web collection on statistics for biologists contains articles on many of the points above.*

## Software and code

Policy information about availability of computer code

| | |
|---|---|
| Data collection | ASIO by Steinberg Media Technologies GmbH was used to record multichannel audio files |
| Data analysis | - Final Cut Pro X (v 10.4.8) by Apple Inc. was used to synchronize video files<br>- Tracker (v 5.1.4) by Open Source Physics was used to track the birds' positions in the footage<br>- Audacity (v 2.4.2) by Audacity Team was used to analyse sound files<br>- Matlab (v 2020a) by The MathWorks was used for statistical analysis |

For manuscripts utilizing custom algorithms or software that are central to the research but not yet described in published literature, software must be made available to editors and reviewers. We strongly encourage code deposition in a community repository (e.g. GitHub). See the Nature Portfolio guidelines for submitting code & software for further information.

## Data

Policy information about availability of data

All manuscripts must include a data availability statement. This statement should provide the following information, where applicable:

- Accession codes, unique identifiers, or web links for publicly available datasets
- A description of any restrictions on data availability
- For clinical datasets or third party data, please ensure that the statement adheres to our policy

Source data for all figures and Extended Data figures are provided with the paper. The raw data generated during this study are available from the corresponding author (SH) upon reasonable request.

# Field-specific reporting

Please select the one below that is the best fit for your research. If you are not sure, read the appropriate sections before making your selection.

☒ Life sciences  ☐ Behavioural & social sciences  ☐ Ecological, evolutionary & environmental sciences

For a reference copy of the document with all sections, see nature.com/documents/nr-reporting-summary-flat.pdf

# Life sciences study design

All studies must disclose on these points even when the disclosure is negative.

| | |
|---|---|
| Sample size | sample sizes were chosen in a way that they allow for statistical testing of observed effects |
| Data exclusions | vocalization data from one female bird was excluded as the data quality did not allow analysis |
| Replication | all experiments were performed with six individuals and equal behavior was observed in the majority of birds |
| Randomization | all experiments (incl. controls) were performed with all animals |
| Blinding | blinding during data analysis was not possible because experimental conditions were clearly discernible from the raw data |

# Reporting for specific materials, systems and methods

We require information from authors about some types of materials, experimental systems and methods used in many studies. Here, indicate whether each material, system or method listed is relevant to your study. If you are not sure if a list item applies to your research, read the appropriate section before selecting a response.

### Materials & experimental systems

| n/a | Involved in the study |
|---|---|
| ☒ | ☐ Antibodies |
| ☒ | ☐ Eukaryotic cell lines |
| ☒ | ☐ Palaeontology and archaeology |
| ☐ | ☒ Animals and other organisms |
| ☒ | ☐ Human research participants |
| ☒ | ☐ Clinical data |
| ☒ | ☐ Dual use research of concern |

### Methods

| n/a | Involved in the study |
|---|---|
| ☒ | ☐ ChIP-seq |
| ☒ | ☐ Flow cytometry |
| ☒ | ☐ MRI-based neuroimaging |

## Animals and other organisms

Policy information about studies involving animals; ARRIVE guidelines recommended for reporting animal research

| | |
|---|---|
| Laboratory animals | zebra finch (Taeniopygia guttata), 2 males and 5 females, adult |
| Wild animals | the study did not involve wild animals |
| Field-collected samples | the study did not involve samples collected from the field |
| Ethics oversight | Government of the Free State of Bavaria, Germany |

Note that full information on the approval of the study protocol must also be provided in the manuscript.

