## [Peer Review File · Nature Ecology & Evolution]

Peer Review Information

Journal: Nature Ecology & Evolution

Manuscript Title: Vision and vocal communication guide 3D spatial coordination of zebra finches during wind-tunnel flights

Corresponding author name(s): Susanne Hoffmann

Editorial Notes:

Reviewer Comments & Decisions:

Decision Letter, initial version:
--

5th January 2022

Dear Dr Hoffmann,

Your manuscript entitled "Vision and vocal communication guide 3-D bird flock formation during flight" has now been seen by 3 reviewers, whose comments are attached. The reviewers have raised a number of concerns which will need to be addressed before we can offer publication in Nature Ecology & Evolution. We will therefore need to see your responses to the criticisms raised and to some editorial concerns, along with a revised manuscript, before we can reach a final decision regarding publication.

Given the reviewer concerns regarding statistical and analytical treatments we will need to see special attention devoted to overcoming issues in this regard.

We therefore invite you to revise your manuscript taking into account all reviewer and editor comments. Please highlight all changes in the manuscript text file [OPTIONAL: in Microsoft Word format].

* If you have not done so already please begin to revise your manuscript so that it conforms to our Article format instructions at <http://www.nature.com/natecolevol/info/final-submission>. Refer also to any guidelines provided in this letter.

[REDACTED]

Nature Ecology & Evolution is committed to improving transparency in authorship. As part of our efforts in this direction, we are now requesting that all authors identified as 'corresponding author' on published papers create and link their Open Researcher and Contributor Identifier (ORCID) with their account on the Manuscript Tracking System (MTS), prior to acceptance. ORCID helps the scientific community achieve unambiguous attribution of all scholarly contributions. You can create and link your ORCID from the home page of the MTS by clicking on 'Modify my Springer Nature account'. For more information please visit please visit www.springernature.com/orcid.

[REDACTED]

Reviewer expertise:

Reviewer #1: movement ecology and aeroecology

Reviewer #2: collective behaviour in flocking birds

Reviewer #3: physics of collective animal behaviour

Reviewers' comments:

Reviewer #1 (Remarks to the Author):

I really enjoyed reading this study. The findings and the approach are very novel, with authors using cameras and animal attached microphones on zebra finches flying in a wind tunnel to examine how birds orientate with respect to the test section walls and to each-other, when flying in a flock of 6. The flight durations are short, but this is fairly typical of wind tunnel studies, and the use of the wind tunnel has enabled the novel insight provided here, as the authors describe in the discussion.

This is a results-heavy paper, with many interesting findings that are glossed over because they are considered in the results section. Separating things into a more traditional Results and Discussion would help to reduce some of the descriptive detail, and give more airtime to the interpretation of the results. I also think it would be good to give more consideration of when collisions with other birds are likely in a general sense, in order to provide more context about the selective pressure for collision avoidance mechanisms. All my comments can be easily addressed.

Results, P 4, lines 22 – it might be of interest that the increased variability of bird position when flying solo is consistent with that seen in homing pigeons as reported by Garde et al 2020, Royal Society Open Science

P5, line 14, it's hard to tell whether these average figures are meaningful without an indication of the variance

Fig. 1. It looks as though there is only a partial field of view of the cameras from panels B and C. How much of the flight time were birds in view of 1 or both cameras?

P7, lines 17-18, it would be clearer to give the number of collisions and the total flight time for group flights. The absolute number of collisions is also a point of interest

P11, it would be useful to have more information about the body of evidence about the function of Distance and Stack calls (ref 15)

P 11 line 17, it could also be beneficial to conserve energy during flight, due to the high metabolic costs

3P13, first paragraph. Is the transparency of the left panel and the ability to see the aviary relevant to the birds calling less from this position? I don't see how this paragraph follows from the previous one. – Ok, so the explanation comes on p15, but it's disconnected from this descriptive information.

P13, line 10, to what extent were the stack calls in sustained flight produced by one individual? Can you break down the number of calls across individuals and contexts (i.e. positions) within sustained flights?

P15, based on the information about the lateralisation of vision in these birds, are there any other positional changes that you would predict might be accompanied by a call? I guess I'm asking about the general rules that you can extract from your results.

Discussion.

(1) I found the format rather odd, having 2 paragraphs of very general conclusions as a discussion, with such a long results section. I guess that this is a hangover from submission to another journal and the editor will advise here, but I would have found a more traditional results and discussion section more helpful, as at the moment the reader is walked through a lot of results at the same time as piecing the interpretation together. Some of the detail in the description of the results could be reduced.

(2) While the authors point out that it's unknown whether their findings would transfer to other species or scenarios I would still value more information on (i) the extent to which other birds call when flying in a flock and (ii) some consideration of whether their results could provide insight into general mechanisms e.g. based on lateralisation of vision (how common is this in birds?) and when birds might be more prone to collision with flock mates.

Reviewer #2 (Remarks to the Author):

This is a great manuscript, that shows real innovation to bring together these techniques to address an otherwise inaccessible question. I was very impressed with the figures, which I felt did a fantastic job of conveying your findings in a very clear way. The way that the results section is structured to gradually build up the picture for the reader is also excellent, and I found that every time I started to have a question begin to nag at me, it was almost immediately addressed in the following lines. I have a few suggestions of where clarity might be added, and also a few additional questions outlined below.

Why were only 4 of 13 flight sessions analysed? Were the other 9 the low-level light sessions? If you could break down what these sessions were, that would be most helpful. (Social/Solitary, Noise masked, low light conditions etc). At present I am unable to get a clear impression of how many of each different trial type were used. Potentially a table in supplementary materials outlining the numbers of each, with instances where data was excluded outlined, would be the easiest way to communicate this?

Is it possible to add a graph of the data from the asocial vs social horizontal positional changes to the

4supplementary materials? The numbers provided are quite close together, so it would be helpful to see the data plotted. Also when using the term Hz it would be beneficial to spell out clearly that what this means – a median frequency of horizontal direction changes of 0.4 per second?

Fig 1 conveys lots of info really well, in all parts. It also seems to indicate something akin to individual 'flight signatures'. Were these, and the positioning of the individuals consistent across trials? Was the variation in f (Hz) in the wind direction linked to position in the group also, with higher variation being seen in birds that occupied a position in the middle of the group? My assumption would be that flying centrally comes with the cost of higher levels of flight adjustment as a result of having birds both ahead and behind.

In the section addressing vocalisations, the frequency of vocalisation once flying is only a little higher than the frequency of collisions. The authors state that the low calling rate once flight is established is likely to be an adaptation so as not to draw the attention of predators to the group, but to my mind the frequency with which the group would encounter a predator that is in a sufficiently advantageous position to launch an attack would be rare. A falcon on the wing would be clearly obvious against the sky, while a perched raptor would need to be in a position higher than the finches, with the frequency of refuges for the finches at a low enough level to make the commencement of an attack worthwhile. Maybe a cat or similar could be alerted to their approach by the calls and leap to strike one from the sky, but it seems unlikely. Might the cognitive load and need to prioritise focus on processing visual information be taking priority? The wind tunnel appears to present a very challenging environment in which to flock, and while calls are especially important initially, might the drop off in their use be because continuing to call at that frequency is unsustainable for long periods of time, and this is only increased when necessity (e.g. low light levels) dictates? If collisions do occur quite frequently, surely these present as great a danger to the birds as predators do, especially given that collisions create instances where the individuals involved become especially vulnerable to being targeted by any predator attack (due to oddity effects, and loss of cohesion with the flock)?

What impact of take-off calls on group order and organisation? This seems likely to be the most data-rich area of the study, but is largely ignored. Did you look into this at all? The approach being employed in this experiment is, to my knowledge, very novel, so it is hardly surprising that the paper does not focus on the area where it was perhaps more straightforward to unpick (plus it is already a very comprehensive and complete manuscript!). Nevertheless, it would be good to have this addressed at some point, even just if it is only to highlight this as the next area for investigation in the discussion.

Following from my previous point, given that take-off is when the maximum level of calling occurs, can you venture any explanations for why collision rate did not increase during the trials with low light levels and masking noise?

This last point is probably my most important. You mention in Page 13 line 9 that bird Green is responsible for most of the vocalisations in the parts of the study he was alive for, and that this is consistent with the typical position of this bird in the flock. But could you not be confusing cause with effect? Might your results be driven by the fact that your most vocal bird just happened to typically fly in this location? Throughout the manuscript you use analyses that do not take into account that the

5data is derived from multiple trials involving the same individuals. This may place constraints upon how generally applicable your results are, a point that is given some attention in the discussion (Page 22, line 9). At present there is no justification given for why you used certain comparison types, despite the fact that these simple contrasts do not account for repeated measures of the same individuals and the pseudoreplication that exists. To my eyes, this is the main barrier to the paper being publishable in its current form.

I also have the line-specific queries outlined below:

Page 3 Line 7: Suggest changing to 'adapt their own movements'

Page 4 Line 6: It would be valuable to highlight that Hz refers to the number of cycles per second, as this may be unclear to a reader who is unfamiliar with how it appears to apply to so many contexts. Simple enough to google, I know, but the easier to read and digest the paper is the wider the appeal, and I feel this manuscript may be of great interest to readers for whom collective behaviour is not their main avenue of research.

Page 5 Line 9: You could aid clarity here by summarising that birds at the front of the flock stayed at the front of the flock, and that those in the middle/back did likewise.

Page 5 Line 20: In this section alignment and misalignment are reported, but without specification of what the cut off for this is. If bird A is at angle 0, how many degrees in bearing does bird B need to be at, and for how long, before is it considered to be misaligned? Does it ride on the correlation coefficient being above 0.3?

Page 6 Line 2: Following on from my previous point, does this period of alignment fall outside what would be expected by chance? It only requires a momentary alignment of 4 individuals, and surely in the midst of all horizontal weaving there might be some moments without that would make this alignment more possible. Was there anything that might explain why these brief moments of cohesive alignment were detected? The analysis of this part appears to serve the purpose of providing evidence that the flight of the group is truly a case of collective behaviour, and not 6 birds flying asocially in a wind tunnel because they have little alternative. Given that they cannot point in directions other than into the wind, and cannot make turns of any great size without being blown away, I can't help but feel you might be applying too strict a criterion for alignment?

Page 7 Line 18: A collision rate of 0.02 Hz nevertheless equates to one every 50 seconds (unless I am calculating incorrectly), which seems high. Having never used wind tunnels myself, how consistent are the conditions they generate? In the videos the movement of the birds suggests they may be variably, or have unpredictable wind eddies/oscillations, is this the case, and might it explain why the collisions occur so frequently?

Page 7 Line 24: This contrasts with the opening lines (Page 4 Line 6), where you state that you samples the video recordings at 24Hz. To avoid confusion, it may be helpful that you subsampled at a 24Hz rate from the original 120Hz recordings?

Page 19 Line 5: Can you add the number of trials that you performed for each of the low light conditions? I imagine it is in the methods, but stating it here saves the reader from having to bounce back and forth between different parts of the paper. Or create a table outlining the number of sessions that were performed and analysed of each type in the Supp Mat, as per my earlier suggestion.

Page 22 Line 9: 'Whether' rather than 'If', would probably be a better opening to this sentence.

Figure S3: Can you make the red dots in this plot slightly larger? I had difficulty making them out unless I zoomed in to 200%. Adding a notation to indicate the wind direction in the still frame would also help to add clarity.

Reviewer #3 (Remarks to the Author):

In the manuscript titled "Vision and vocal communication guide 3-D bird flock formation during flight" the authors report a collective behaviour study of flight coordination based on high resolution tracking and acoustic recording to understand the importance of visual and vocal communication. They performed experiments with small groups of zebra finches (groups of 6 individuals) flying in a large wind tunnel. By analysing the measured flight trajectories and the acoustic recordings, their study confidently and elegantly shows that the birds use both visual information and active vocal signalling to coordinate their movement path in order to avoid collisions. I find the paper useful and interesting, and the topic may fit well the interest of the readers of the journal. The manuscript is generally well written and it is clear to follow. Performing such experiments and analysing those data are very hard tasks that provide several challenges in relation to training and handling the animals, operating the wind tunnel, getting reliable tracking data, recording individual vocalisation of each bird, using appropriate methods for analysing the data, etc. Overall, the authors did great job in tackling these challenges. That said, I have several comments for which I give a detailed list below.

Major comments:

The title, the first sentences of the abstract, the beginning of the introduction and the last sentence of the discussion give the impression that (i) the observed phenomena could be universal across different taxa performing collective motion and (ii) the observed flight coordination is in relation to the large scale movement decision of a group. I find both of these implicit assumptions incorrect. For the first one (i), just to mention trivial counter-examples there are numerous systems in animal collective motion where the individuals are not even able to communicate using vocal signalling or they don't have vision. But even when narrowing the scope for animal groups in which both the visual perception and the vocal communication exist, there are very different ways how the rules of group coordination could depend on the size, group structure or other factors. For example, in case of very large groups consisting of thousands of birds, such vocal communication could only have local effect. Or groups may have a well-defined structure (being egalitarian, despotic, or multilevel hierarchical, etc.) in contrast to the characteristics of the zebra finch groups. For the second (ii), in collective behaviour studies it is an important question how the movement of the group emerges from the local interactions and the behavioural decisions of the individuals. In the current study, there is no way how

7these local interactions would impact large scale behaviour of the group, as the main direction of flight is forced by the flow of the air in the wind tunnel. The authors themselves report in this study that in most of the occurrences those relatively large changes in the spatial position of an individual have almost no effect on the behaviour of others. So these local movement directional decisions are not propagating in the flock. While I agree that models of collective motion could be extended in the direction to include active vocal signalling, but the implication of the results of the current work is not directly applicable for such studies. So statements related to this should be rephrased to be more precise.

Based on what is written above, I suggest that the title should be more specific about the observed phenomena and the main findings, and the abstract and the introduction should also be corrected. (To be fair I would like to mention that the first two sentences of the discussion point out this limitation clearly, but I think that is not enough).

My other major general concern is about how universal is the observed behaviour even for zebra finch groups. Based on the videos and the tracking data, it is clear that during their flight in the wind tunnel birds are oscillating (constantly changing their relative position to each other - mainly sideways). Does this occur in natural free flying zebra finch flocks? Or could this be an artificial effect (some stress induced or attempted escape reaction) that is caused by flying in the wind tunnel? The authors should include some explanation in relation to this.

In connection to the previous point, it is not clear that how does the typical inter-individual distance in the current wind tunnel experiment relate to typical values for free flying groups of zebra finches. Is density in the wind tunnel in the order of natural free flight or rather densely packed?

The trajectory data analysis is mostly based on detecting the periodic oscillation in the signal by using periodogram function. That should be correct if a periodicity in the signal exists, which is the case for many of the examples given. But if the signal does not have regular changes (because oscillation occurs irregularly or oscillations are not even present), then the values given by that function are rather misleading. The power spectrum will still have a highest value, but reporting that is not very informative. For example, Fig. 1E shows birds with mostly periodic oscillation (horizontal position: purple, yellow, blue) but also shows birds that have irregular oscillations (horizontal position: magenta, black). The periodicity value given by f does not capture whether the oscillation is present or not. Horizontal position: magenta and vertical position: yellow have almost the same f value, and the former has oscillation with a characteristic time duration just happening at irregular times, and the later does not have significant oscillation. Also, if the signal has lower and higher frequency harmonic components, the current method used (i.e. reporting only the highest value of the periodogram) may not give the correct value that actually characterises the motion pattern. This is clear in some of the cases where high frequency oscillation is present in the signal, but still the reported f is very low. I suggest taking into consideration the strength of the power spectrum as well to characterize whether the periodicity is captured by f or not. Unfortunately, this would mean that all the statistics that use f as the main value may need to be recalculated.

For some of the reported statistics, it is not clear what was used as "data points". There are several places where n is not reported (if there is some common value for all of them that still should be

clearly stated somewhere). Were only independent points used for the statistics, so no artificial inflation of number of data points occurs? How was that assured? Could the authors comment on that?

The authors use correlation to detect coordinated movement within the flock. The individuals perform motion patterns composed of multiple spatial and temporal scales. It is correct to check correlated movement in the co-moving coordinate system as done by using tracking data in the wind tunnel. But I think the description gives a misleading impression that during most parts of the flight the birds move in an uncorrelated way. In contrast to this, the main flight direction is same for all birds so their motion is very much correlated, as the main direction of the motion is forced by the air flow in the wind tunnel. I suggest to clarify this at around the first paragraph of page 6.

I really liked the experiments with low light intensity and with additional noise to mask the calls. The results of these experiments strongly indicate the importance of using specific information to avoid collisions.

Minor comments:

Maybe I missed, but I did not find the definition about the meaning of the spatial coordinates. On the figure "arbitrary unit" is being used, I guess that the arbitrary unit relates to the pixel coordinates provided by the video tracking. If so, the values reported are strongly affected by parallax and perspective effects. Although the current analyses (based on power spectrum, for example) may not be affected by that (which is a clever choice of the authors), I think it would necessary to include some explanation for this.

Maybe I missed this as well, but I did not find (and I found puzzling) where does the value of 209ms originate from (Page 13 Line 16) in relation to Stack calls.

Page 13 Line 22: The authors state that based on the non-significant Rayleigh test the distribution is uniform. This is problematic and I suggest rewriting this sentence, as the values which are shown on Fig. 3D are far from being uniformly distributed. The Rayleigh test could show if a circular data has a unimodal peak or not, but here the data is rather inhomogeneous having two peaks on both sides. That could cause the observed non-significant output of the test. So I think there should be a more appropriate method used (for example Hermans-Rasson test?). This is true for the other similar plots, although because of the large number of data point, those could be significant using Rayleigh test even for the bimodal cases (shown on panels Fig. 3F - right side).

The figures are generally nicely structured and neat. One improvement that I would suggest is that the readability could be a bit better supported by using more panel labels and/or text labels on the panels where there are multiple plots belonging to a single panel label (instead of just writing in the legend left/right and top/bottom).

Page 15 Line 3: I think the value given there has a typo, as the average of the 4 numbers given in the bracket is 10.25 and not 10.025. Moreover (very minor comment), I doubt that the precision of the value is accurate to the 2nd decimal anyways, so I think it would be more readable to give as 41:540 reporting simply the sum rather than the mean of the values.

9Fig. S2. (position in wind direction plots, for example the top right group) here several positions are outside of the given range. Is this a results of the plotting, or the tracking is not available for these parts? How do such missing points impact the periodogram analysis?

Fig. S7. It is not clear what is shown on the bottom plot?

Table S1 (and Page 5 Line 14 for example) Why is there a need for using a relative score that is divided by the maximum distance? This makes these relative values very sensitive to large outliers.

Summary

As a consequence of all the above, I suggest a revision, although I think the changes will not impact the main conclusions of the paper. I would be happy to review the improved version of the manuscript.

*****END*****

Author Rebuttal to Initial comments

Reviewers' comments:

Reviewer #1 (Remarks to the Author):

I really enjoyed reading this study. The findings and the approach are very novel, with authors using cameras and animal attached microphones on zebra finches flying in a wind tunnel to examine how birds orientate with respect to the test section walls and to each-other, when flying in a flock of 6. The flight durations are short, but this is fairly typical of wind tunnel studies, and the use of the wind tunnel has enabled the novel insight provided here, as the authors describe in the discussion.

Thank you for the positive and encouraging assessment of our manuscript and for the helpful comments and suggestions.

This is a results-heavy paper, with many interesting findings that are glossed over because they are considered in the results section. Separating things into a more traditional Results and Discussion would help to reduce some of the descriptive detail, and give more airtime to the interpretation of the results. I also think it would be good to give more consideration of when collisions with other birds are likely in a general sense, in order to provide more context about the selective pressure for collision avoidance mechanisms. All my comments can be easily addressed.

Especially because our manuscript combines many different results, we feel that providing an interpretation for a certain finding and framing the next question that arises from this interpretation before transitioning to the next experiment in the Results section, gives the manuscript a better flow and may facilitate understanding in the reader. However, we understand your concern and asked for editorial advice. The editor replied that Nature Ecology and Evolution frequently publishes research articles in the format of our manuscript, and because the other two reviewers seem to be fine with the manuscript's current structure, the editor recommended to retain the structure.

Various studies have been published that report on collisions between birds and man-made structures, such as buildings or wind turbines. However, to the best of our knowledge, we are not aware of any study investigating collisions between individual birds within a flock. Our experiments suggest that zebra finches collide more often with each other during flight when they are not able to communicate vocally, but since there is no data from other species that support our findings, we do not feel confident to draw general conclusions based on our results. Please also see our reply to your last comment.

Results, P 4, lines 22 – it might be of interest that the increased variability of bird position when flying solo is consistent with that seen in homing pigeons as reported by Garde et al 2020, Royal Society Open Science

Thanks, this is an interesting study, which we were not aware of. We added this information to the manuscript (page 5 lines 17-19).

P5, line 14, it's hard to tell whether these average figures are meaningful without an indication of the variance

Yes, thanks. The revised Fig. 1f now displays mean values and standard deviations.

Fig. 1. It looks as though there is only a partial field of view of the cameras from panels B and C. How much of the flight time were birds in view of 1 or both cameras?

In Figures 1b and c, cutouts of the freeze frames are displayed to facilitate detection of the birds by the reader. While all birds were in the field of view of Camera 1 from start to end of all flight sessions, birds were in the field of view of Camera 2 only during the sustained phase of flight, and were able to transiently exit the field of view of Camera 2 when flying at very upwind or downwind, or at very lateral positions in the flight section. For the four flight sessions in which we tracked the birds' positions throughout the entire session, a bird was on average for 77.1% of the time in the field of view of Camera 2. This information is now provided in the manuscript (page 29 lines 11-14).

P7, lines 17-18, it would be clearer to give the number of collisions and the total flight time for group flights. The absolute number of collisions is also a point of interest

Flight sessions were variable in duration. To allow comparison of collision probability between sessions performed under different experimental conditions, we decided to calculate collision rates instead of indicating the number of collisions. However, we added information on total flight time and total number of collisions to the new version of the manuscript (page 8 line 16).

P11, it would be useful to have more information about the body of evidence about the function of Distance and Stack calls (ref 15)

Except for alarm calls, the function of call types is poorly studied in birds. For the Stack call, the book by Richard Zann (ref. #16) is, to the best of our knowledge, the only publication providing information about this call's function. For the Distance call, we added a second reference (ref. #27, Elie & Theunissen, 2016, Anim Cogn 19:285–315) to the new version of the manuscript (page 12 line 8). This study describes the behavioral context in which Distance calls are emitted by captive zebra finches, replicating the findings of Zann, 1996 in wild birds.

P 11 line 17, it could also be beneficial to conserve energy during flight, due to the high metabolic costs

While during flight zebra finches expend 27.8 times more energy than during non-flying periods (Nudds & Bryant, 2000, J Exp Biol 203(Pt 10):1561-72), during song production energy expenditure is increased by only 1.2 – 1.5 times in this species (Franz & Goller, 2003, J Exp Biol 206(Pt 6):967-78). When considering that song production takes between 1-4 seconds and call production only 50-300 ms, the metabolic cost of calling should be negligible and we therefore doubt that reducing metabolic cost is a main cause for the reduction of call emission rate during sustained flight in zebra finches.

P13, first paragraph. Is the transparency of the left panel and the ability to see the aviary relevant to the birds calling less from this position? I don't see how this paragraph follows from the previous one. – Ok, so the explanation comes on p15, but it's disconnected from this descriptive information.

No, we do not expect that the bird's ability to see the aviary influences the calling activity. However, we suggest that the transparency of the left side wall may explain the birds' tendency to fly left from the center of the flight section. We now recognize that the

2

paragraph describing the transparency of the flight section's walls does not fit well into the text at the former position and moved it to the beginning of the results section where we describe the general flight behavior (page 4 lines 16-20).

P13, line 10, to what extent were the stack calls in sustained flight produced by one individual? Can you break down the number of calls across individuals and contexts (i.e. positions) within sustained flights?

Bird Green called most frequently (N = 60 calls), probably because he was mainly flying at the lower edge of the flock and therefore often in a position where he was not able to see all other members of the flock when moving upwards (page 14 line 1-3). Bird Light blue emitted 16 calls, bird Lilac 9 calls, bird Pink 7 calls and bird Orange 1 call. This information is now also provided in the caption of Extended Data Fig. 7.

P15, based on the information about the lateralisation of vision in these birds, are there any other positional changes that you would predict might be accompanied by a call? I guess I'm asking about the general rules that you can extract from your results.

From the flight sessions we performed under low light conditions, we know that zebra finches increasingly use vocal communication when visual input is not reliable. Therefore, we suggest that when flying under normal light conditions they use vocal communication when moving towards spatial positions where they do not receive visual input from, such as positions above or behind themselves. Zebra finches and also most other birds are not able to fly backwards. Consequently, upwards directed movements might represent the only type of positional change during which a zebra finch flying in a flock under normal light conditions in open space might make use of vocal communication. Hypothetically, vocal communication can also be beneficial for bird flocks flying during the night or through foggy or cloudy air. However, this is only speculation.

Discussion.

(1) I found the format rather odd, having 2 paragraphs of very general conclusions as a discussion, with such a long results section. I guess that this is a hangover from submission to another journal and the editor will advise here, but I would have found a more traditional results and discussion section more helpful, as at the moment the reader is walked through a lot of results at the same time as piecing the interpretation together. Some of the detail in the description of the results could be reduced.

We addressed the content of this comment already at the beginning of this response letter and provide here only a brief recap: although we understand your concern, we feel that providing an interpretation for a certain finding and framing the next question that arises from this interpretation before transitioning to the next experiment in the Results section gives our manuscript a better flow. Following editorial advice, we decided to retain the current structure of the manuscript.

(2) While the authors point out that it's unknown whether their findings would transfer to other species or scenarios I would still value more information on (i) the extent to which other birds call when flying in a flock and (ii) some consideration of whether their results could provide insight into

general mechanisms e.g. based on lateralisation of vision (how common is this in birds?) and when birds might be more prone to collision with flock mates.

Many bird species, especially migratory species, emit calls, named “flight calls”, during flight. The function of these calls, however, is still largely unknown. There is only anecdotal evidence that flight calls may aid in group maintenance (Farnsworth, 2005, The Auk 122(3):733-746). This information is now provided at page 3 lines 11-14 in the revised manuscript. Visual lateralization seems to be conserved amongst avian species. With cognitive, pharmacological or physiological experiments, it has been shown in several species that the left and the right hemisphere of the avian brain process to some extent different aspects of visual information. In our manuscript, we suggest visual lateralization may be one of the reasons for zebra finches flying at specific positions in the flock to not see their conspecifics and therefore to emit calls before moving upwards. This is only a hypothesis, which cannot be directly tested with our experiments. Thus, we do not feel confident to draw conclusions regarding general mechanisms based on our results. Furthermore, we tested only one bird species, and comparative data from other species do not exist. Therefore, the generalizability of our findings is unclear (as also mentioned by Reviewer #3). Various studies have been published that report on collisions between birds and man-made structures, such as buildings or wind turbines. However, to the best of our knowledge, we are not aware of any study investigating collisions between individual birds within a flock. Our experiments suggest that zebra finches collide more often with each other during flight when they are not able to communicate vocally, but again, there is no data from other species supporting our findings.

Reviewer #2 (Remarks to the Author):

This is a great manuscript, that shows real innovation to bring together these techniques to address an otherwise inaccessible question. I was very impressed with the figures, which I felt did a fantastic job of conveying your findings in a very clear way. The way that the results section is structured to gradually build up the picture for the reader is also excellent, and I found that every time I started to have a question begin to nag at me, it was almost immediately addressed in the following lines. I have a few suggestions of where clarity might be added, and also a few additional questions outlined below.

Thank you for the positive and encouraging assessment of our manuscript, and for the helpful and constructive comments.

Why were only 4 of 13 flight sessions analysed? Were the other 9 the low-level light sessions? If you could break down what these sessions were, that would be most helpful. (Social/Solitary, Noise masked, low light conditions etc). At present I am unable to get a clear impression of how many of each different trial type were used. Potentially a table in supplementary materials outlining the numbers of each, with instances where data was excluded outlined, would be the easiest way to communicate this?

In the footage of both cameras, we tracked the positions of the birds manually, a method that requires a tremendous amount of time. To describe the general flight behavior of the birds in the flight section when flying at normal light conditions, we initially tracked the birds' positions throughout four entire flight sessions (session #2, #5, #8, #13). The four sessions were chosen so that they were more or less equally distributed throughout the time of the experiment. This rationale is included in the Methods section of the manuscript (page 31 last paragraph). After recognizing that the birds' flight behavior was similar throughout these four sessions, we decided not to analyze the birds' flight behavior in the remaining sessions. However, please note that only for the description of the general flight behavior, analysis was restricted to four flight sessions. Analyses of vocal behavior and correlation of vocal and flight behavior was performed on all sessions. Providing an overview of the numbers of performed/analyzed flight sessions for each flight condition in a table format, is an excellent suggestion. Such a table is now provided in the supplementary material (Supplementary Table 2).

Is it possible to add a graph of the data from the asocial vs social horizontal positional changes to the supplementary materials? The numbers provided are quite close together, so it would be helpful to see the data plotted. Also when using the term Hz it would be beneficial to spell out clearly that what this means – a median frequency of horizontal direction changes of 0.4 per second?

Yes, sure. We added the new Extended Data Fig. 4 to the manuscript. This figure displays all frequencies of position change determined for all birds in four flight sessions performed in flocking flight and solo flight, respectively. Yes, a median position change frequency of 0.4 Hz means a median frequency of direction change of 0.4 per second. This is now explained at page 5 line 12.

Fig 1 conveys lots of info really well, in all parts. It also seems to indicate something akin to individual 'flight signatures'. Were these, and the positioning of the individuals consistent across trials? Was the variation in f (Hz) in the wind direction linked to position in the group also, with higher variation

being seen in birds that occupied a position in the middle of the group? My assumption would be that flying centrally comes with the cost of higher levels of flight adjustment as a result of having birds both ahead and behind.

Yes, both the pattern of positional changes and the area within the flight section a bird preferred to fly in seemed to be consistent across the four flight sessions we analyzed. Bird Green always preferred to fly at low positions in the flight section, bird Lilac preferred to fly at frontal positions and birds Black, Light blue and Orange preferred to fly at rear positions of the flock. Bird Pink was most often located in the middle part of the flock. This can nicely be seen in Fig. 1d and Extended Data Fig. 1, and is mentioned at page 4 lines 12-15. In all four analyzed flight sessions, bird Lilac cycled between the left and right side of the flight section, while all other birds tended to spend more time on the left side before moving to the right side of the flight section. And after moving to the right, all birds moved back to the left side immediately. Although individual "flight signatures" seem to exist, the frequency of position changes doesn't seem to depend on a bird's preferred position in the flock. In three of the four sessions, bird Pink (flying in the middle of the flock) showed lower horizontal position change frequencies than bird Lilac (flying at the front of the flock).

In the section addressing vocalisations, the frequency of vocalisation once flying is only a little higher than the frequency of collisions. The authors state that the low calling rate once flight is established is likely to be an adaptation so as not to draw the attention of predators to the group, but to my mind the frequency with which the group would encounter a predator that is in a sufficiently advantageous position to launch an attack would be rare. A falcon on the wing would be clearly obvious against the sky, while a perched raptor would need to be in a position higher than the finches, with the frequency of refuges for the finches at a low enough level to make the commencement of an attack worthwhile. Maybe a cat or similar could be alerted to their approach by the calls and leap to strike one from the sky, but it seems unlikely. Might the cognitive load and need to prioritise focus on processing visual information be taking priority? The wind tunnel appears to present a very challenging environment in which to flock, and while calls are especially important initially, might the drop off in their use be because continuing to call at that frequency is unsustainable for long periods of time, and this is only increased when necessity (e.g. low light levels) dictates? If collisions do occur quite frequently, surely these present as great a danger to the birds as predators do, especially given that collisions create instances where the individuals involved become especially vulnerable to being targeted by any predator attack (due to oddity effects, and loss of cohesion with the flock)?

When perched in trees or bushes in their natural habitat, zebra finches are very noisy. They are basically calling all the time. While hidden in bushes, the spatial positions of individual finches are likely difficult to determine for raptors on the wing or perched in a nearby tree. However, when flying, calls can inform predators about a flock of finches passing by, and without the protective visual clutter generated by leaves and branches, the positions of single individuals can be easily detected by a raptor. Therefore, we felt that choosing to be less conspicuous by reducing the frequency of call emissions might be a good explanation for the decrease in call rates after take-off.

When hopping around within a bush, the finches have to visually estimate distances between branches while they simultaneously receive visual information from conspecifics. In our opinion, the processing load of visual information is not much smaller when perched in a bush than when flying in a flock, and we would therefore argue against a shift in sensory focus as an explanation for the decrease in calling rates during flight.

6

16We also would exclude the hypothesis that the finches reduce calling rates to save energy. While during flight, zebra finches expend 27.8 times more energy than during non-flying periods (Nudds & Bryant, 2000, J Exp Biol 203(Pt 10):1561-72), during song production energy expenditure is increased by only 1.2 – 1.5 times in this species (Franz & Goller, 2003, J Exp Biol 206(Pt 6):967-78). When considering that song production takes between 1-4 seconds and call production only 50-300 ms, the metabolic cost of calling should be negligible and we therefore doubt that reducing metabolic cost is a main cause for the reduction of call emission rate during sustained flight in zebra finches.

From our observations in the wind tunnel, we know that mid-air collisions don't seem to be physically harmful to zebra finches. When they bump into each other, they struggle for some milliseconds but then easily find back into their flight rhythm without losing much flight height or speed. But yes, collisions could create disturbances in a flock, which might be conspicuous to a raptor and might make single individuals more vulnerable to an attack. But this holds only true when the raptor had already detected the flying flock. And sound emission as an omnidirectional, long-distance signal can call a predator's attention towards the flying flock even when they were not aware of the flock before. Consequently, we still think that being less conspicuous is the best reason for the zebra finches to reduce the frequency of call emission during flight, and to call only in situations where vocal communication is inevitable.

What impact of take-off calls on group order and organisation? This seems likely to be the most data-rich area of the study, but is largely ignored. Did you look into this at all? The approach being employed in this experiment is, to my knowledge, very novel, so it is hardly surprising that the paper does not focus on the area where it was perhaps more straightforward to unpick (plus it is already a very comprehensive and complete manuscript!). Nevertheless, it would be good to have this addressed at some point, even just if it is only to highlight this as the next area for investigation in the discussion.

Yes, we did not analyze the birds' flight behavior at the time of call emission during the take-off phase. The main question we wanted to answer with our study was: Do birds use vocal communication for flock coordination during flight? During flight sessions in the wind tunnel, birds never took off all together but one after another with temporal delays of up to one second between birds. Therefore, when a bird called when taking off from the perch, some flock mates were already flying and others were still sitting on the perch. The whole flock was consistently flying only after a few seconds after the first bird's take-off. Consequently, to investigate the behavior of the whole group during flight, we restricted our analysis to the period of the flight sessions where all birds were indeed flying. We also did not analyze distance calls, which were also (but only very rarely) emitted during sustained flight. This is now mentioned in the revised version of the manuscript (page 13 lines 13-16).

Following from my previous point, given that take-off is when the maximum level of calling occurs, can you venture any explanations for why collision rate did not increase during the trials with low light levels and masking noise?

This is an intriguing question, however, we never performed flight sessions at a low light level and in the presence of masking noise at the same time. We suggest that collision rates did not increase with decreasing light level because the birds were still able to use calls to communicate imminent position changes to their flock mates. If we would have combined low light levels with the presentation of masking noise, we likely would have seen an increase in collision rates. But this is purely hypothetical.

7

17This last point is probably my most important. You mention in Page 13 line 9 that bird Green is responsible for most of the vocalisations in the parts of the study he was alive for, and that this is consistent with the typical position of this bird in the flock. But could you not be confusing cause with effect? Might your results be driven by the fact that your most vocal bird just happened to typically fly in this location? Throughout the manuscript you use analyses that do not take into account that the data is derived from multiple trials involving the same individuals. This may place constraints upon how generally applicable your results are, a point that is given some attention in the discussion (Page 22, line 9). At present there is no justification given for why you used certain comparison types, despite the fact that these simple contrasts do not account for repeated measures of the same individuals and the pseudoreplication that exists. To my eyes, this is the main barrier to the paper being publishable in its current form.

Yes, when only considering our finding that birds flying at low positions in the flock are more likely to emit a call, it seems to be problematic that bird Green, which emitted the majority of the vocalizations, prefers to fly at very low positions. When a highly vocally active bird prefers to fly at a certain position in the flock, this could indeed be confused with the assumption that this position in the flock drives the bird to vocalize. However, in the following paragraph in the manuscript, we show that calling is not only related to the calling bird's spatial position within the flying flock but also to its flight movements. A bird emits a call when it is located at the bottom edge of the flock AND intends to move upwards. We clarify this at page 17 lines 4-10. When only increased vocal activity would drive the bird to vocalize when flying at low positions in the flock, we would not see a clear correlation between calling events and specific movement (i.e. upwards directed position changes). Therefore, we don't think that our results are affected by individual differences in vocal activity between birds.

Training zebra finches to fly in the wind tunnel takes some time, and not every bird is able to perform the task. Therefore, we chose a repeated measure design for our study, and collected data only from one small flock of six zebra finches. This rationale is now provided in the Methods section (page 34 lines 14-16).

However, we agree that not accounting for the repeated measures in our analysis and statistics is a very critical point. Therefore, except for the analysis of circular data, we repeated all statistical analysis, and now use Linear mixed models to describe interactions between measurements. In these models, the individual birds' IDs and the type of repetition, such as session number, were implemented as random effects. Furthermore, we now present individual data in addition to population effects. This allows the reader to judge the contribution of each individual to the population effect.

We also refuse to draw conclusions about the general rules that could be extracted from our results (please see our replies to comments made by Reviewer #1), and made all general statements in the revised manuscript specific to zebra finches (as requested by Reviewer #3).

I also have the line-specific queries outlined below:

Page 3 Line 7: Suggest changing to 'adapt their own movements'

Thanks, we changed this accordingly.

Page 4 Line 6: It would be valuable to highlight that Hz refers to the number of cycles per second, as this may be unclear to a reader who is unfamiliar with how it appears to apply to so many contexts.

Simple enough to google, I know, but the easier to read and digest the paper is the wider the appeal, and I feel this manuscript may be of great interest to readers for whom collective behaviour is not their main avenue of research.

We added the definition of "Hz" at page 4 line 6.

Page 5 Line 9: You could aid clarity here by summarising that birds at the front of the flock stayed at the front of the flock, and that those in the middle/back did likewise.

This would not be entirely true. Birds also changed their position in the flock in wind direction, but in contrast to the other two spatial dimensions they did it only occasionally so that no rhythmicity of position change could be detected. Furthermore, birds were able to exit the field of view of Camera 2 and therefore data on a bird's position in wind direction could be noncontinuous throughout a flight session. Reviewer #3 pointed out that this could have influenced our calculation of position change frequency in wind direction, and we therefore removed this part of the text from the manuscript. An explanation for why we did not determine the position change frequency in wind direction is provided at page 32 lines 9-12.

Page 5 Line 20: In this section alignment and misalignment are reported, but without specification of what the cut off for this is. If bird A is at angle 0, how many degrees in bearing does bird B need to be at, and for how long, before is it considered to be misaligned? Does it ride on the correlation coefficient being above 0.3?

We did not define a threshold for alignment/misalignment. A positive coefficient for the correlation of movement direction between two individuals indicates alignment and a negative correlation coefficient indicates misalignment at a certain point in time. This is now clarified at page 6 lines 6-8 and lines 15-16.

Page 6 Line 2: Following on from my previous point, does this period of alignment fall outside what would be expected by chance? It only requires a momentary alignment of 4 individuals, and surely in the midst of all horizontal weaving there might be some moments without that would make this alignment more possible. Was there anything that might explain why these brief moments of cohesive alignment were detected? The analysis of this part appears to serve the purpose of providing evidence that the flight of the group is truly a case of collective behaviour, and not 6 birds flying asocially in a wind tunnel because they have little alternative. Given that they cannot point in directions other than into the wind, and cannot make turns of any great size without being blown away, I can't help but feel you might be applying too strict a criterion for alignment?

The purpose of the correlational analysis of movement directions was to allow the reader to better understand the internal dynamics in the flock during flight (as suggested during a previous review process with another journal). You are right, the criterion for alignment in the flock is strict, but applies as such to alignment of birds in free flying flocks of storks and pigeons. The fact that there are no extended periods of significant alignment in the zebra finch flock while flying in the wind tunnel therefore just indicates that in a spatially restricted environment and while forced to fly into the wind, zebra finches do not perform collective maneuvers during which movement direction is aligned amongst all birds in the flock. The brief periods of significant alignment we could detect in our recordings highly likely occurred by chance. There was no obvious behavior correlated to these events. This is now explained in the text at page 6 lines 4-6, lines 11-12 and lines 23-24.

9

19Page 7 Line 18: A collision rate of 0.02 Hz nevertheless equates to one every 50 seconds (unless I am calculating incorrectly), which seems high. Having never used wind tunnels myself, how consistent are the conditions they generate? In the videos the movement of the birds suggests they may be variably, or have unpredictable wind eddies/oscillations, is this the case, and might it explain why the collisions occur so frequently?

Our wind tunnel is considered a „Low turbulence“ type, and wind conditions are consistent throughout the flight section. A 6-stage HEXEL-honeycomb rectifier eliminates even the smallest turbulence from the air stream before it enters the flight section, and the air stream can be assumed to be perfectly laminar within the flight section. However, when more than one bird is flying in the flight section, turbulence generated by one bird may influence the flight behavior of another bird.

During an average flight session in normal light conditions with a duration of about one minute, we indeed observed maximally one to two collisions between birds. But considering the spatial limitation experienced by the flock of six birds during flight within the flight section, and the highly dynamic flight style, we actually judged one collision per minute to be rather infrequent. This justification is now included in the manuscript (page 8 lines 13-14). Instead of unpredictable air stream conditions within the flight section, we rather assume the unpredictable flight behavior of conspecifics to cause the collisions.

Page 7 Line 24: This contrasts with the opening lines (Page 4 Line 6), where you state that you samples the video recordings at 24Hz. To avoid confusion, it may be helpful that you subsampled at a 24Hz rate from the original 120Hz recordings?

Both statements are correct. Footage was recorded at 120 frames per second (120 Hz, described in the Methods section at page 29 first paragraph. To investigate the general flight behavior, the birds' positions were tracked in every fifth frame of the footage, resulting in a sample rate of 24 Hz. This is now described in more detail at page 4 line 6. Because the head movements were very fast, we tracked the birds' head positions in every frame of the footage (sample rate 120 Hz, page 8 lines 21-23) to describe the head movements during horizontal position changes.

Page 19 Line 5: Can you add the number of trials that you performed for each of the low light conditions? I imagine it is in the methods, but stating it here saves the reader from having to bounce back and forth between different parts of the paper. Or create a table outlining the number of sessions that were performed and analysed of each type in the Supp Mat, as per my earlier suggestion.

Yes, you are right. It makes sense to mention the number of flight sessions here in the text. This is now done at page 20 lines 20-21. Furthermore, we also included a table (Supplementary Table 2) with an overview of sample sizes.

Page 22 Line 9: 'Whether' rather than 'If', would probably be a better opening to this sentence.

Thanks. We changed this (page 24 line 9).

Figure S3: Can you make the red dots in this plot slightly larger? I had difficulty making them out

unless I zoomed in to 200%. Adding a notation to indicate the wind direction in the still frame would also help to add clarity.

We increased the size of the red dots and explain now in the figure caption (now Extended Data Fig. 5) that wind direction is perpendicular to the plane of the image of the still frame.Reviewer #3 (Remarks to the Author):

In the manuscript titled "Vision and vocal communication guide 3-D bird flock formation during flight" the authors report a collective behaviour study of flight coordination based on high resolution tracking and acoustic recording to understand the importance of visual and vocal communication. They performed experiments with small groups of zebra finches (groups of 6 individuals) flying in a large wind tunnel. By analysing the measured flight trajectories and the acoustic recordings, their study confidently and elegantly shows that the birds use both visual information and active vocal signalling to coordinate their movement path in order to avoid collisions. I find the paper useful and interesting, and the topic may fit well the interest of the readers of the journal. The manuscript is generally well written and it is clear to follow. Performing such experiments and analysing those data are very hard tasks that provide several challenges in relation to training and handling the animals, operating the wind tunnel, getting reliable tracking data, recording individual vocalisation of each bird, using appropriate methods for analysing the data, etc. Overall, the authors did great job in tackling these challenges. That said, I have several comments for which I give a detailed list below.

Thank you for the expert assessment of our manuscript and the helpful and constructive comments and suggestions.

Major comments:

The title, the first sentences of the abstract, the beginning of the introduction and the last sentence of the discussion give the impression that (i) the observed phenomena could be universal across different taxa performing collective motion and (ii) the observed flight coordination is in relation to the large scale movement decision of a group. I find both of these implicit assumptions incorrect. For the first one (i), just to mention trivial counter-examples there are numerous systems in animal collective motion where the individuals are not even able to communicate using vocal signalling or they don't have vision. But even when narrowing the scope for animal groups in which both the visual perception and the vocal communication exist, there are very different ways how the rules of group coordination could depend on the size, group structure or other factors. For example, in case of very large groups consisting of thousands of birds, such vocal communication could only have local effect. Or groups may have a well-defined structure (being egalitarian, despotic, or multilevel hierarchical, etc.) in contrast to the characteristics of the zebra finch groups. For the second (ii), in collective behaviour studies it is an important question how the movement of the group emerges from the local interactions and the behavioural decisions of the individuals. In the current study, there is no way how these local interactions would impact large scale behaviour of the group, as the main direction of flight is forced by the flow of the air in the wind tunnel. The authors themselves report in this study that in most of the occurrences those relatively large changes in the spatial position of an individual have almost no effect on the behaviour of others. So these local movement directional decisions are not propagating in the flock. While I agree that models of collective motion could be extended in the direction to include active vocal signalling, but the implication of the results of the current work is not directly applicable for such studies. So statements related to this should be rephrased to be more precise.

It was never our intention to give the impression that our findings would apply to other taxa or collective behavior systems. Quite the contrary, we actively avoid to draw any general conclusions (please see our replies to comments made by Reviewer #1). However, we agree that the title and the abstract were not specifically phrased for birds/zebra finches. We rewrote the title and added some details to the abstract, to the introduction and to the

12

22discussion to make it more specific to our experiments, and to emphasize that our findings are likely not one-to-one transferrable to large-scale collective behavior.

Based on what is written above, I suggest that the title should be more specific about the observed phenomena and the main findings, and the abstract and the introduction should also be corrected. (To be fair I would like to mention that the first two sentences of the discussion point out this limitation clearly, but I think that is not enough).

We rewrote the title and amended the abstract and the discussion as you suggested. The introduction consists of two parts. In the first paragraph, we introduce the term "collective motion" by mainly citing literature on collective motion in birds, and describe our hypothesis. In the second paragraph, we introduce our animal model, the zebra finch, and briefly mention our most important finding. In the revised version, we added two sentences about flight calls in birds to the first paragraph of the introduction, and exchanged the word "bird" with the word "zebra finches" in the second paragraph. We do not see any other statements here, which are not specific to our study.

My other major general concern is about how universal is the observed behaviour even for zebra finch groups. Based on the videos and the tracking data, it is clear that during their flight in the wind tunnel birds are oscillating (constantly changing their relative position to each other - mainly sideways). Does this occur in natural free flying zebra finch flocks? Or could this be an artificial effect (some stress induced or attempted escape reaction) that is caused by flying in the wind tunnel? The authors should include some explanation in relation to this.

At page 4 line 24 and page 5 lines 1-2, we already addressed the peculiarity of the sideways oscillations, which we observed in birds during flight in the wind tunnel. We wrote that: "This behavior is reminiscent of the flight behavior of wild zebra finches: when being surprised in flight by a predator, zebra finches fly in a rapid zig-zag course low above the ground heading for nearby vegetation (15)." So, yes, sideways position changes during flight seem also to be performed by wild zebra finches. Due to the spatial limitations in the flight section, the birds are forced to fly close to the ground. We suggest that the close proximity to the ground elicits the sideways oscillating flight behavior in both wild, free flying finches and our birds flying in the flight section. However, our hypothesis needs to be proven. We added another sentence to this part of the paper (page 5 lines 2-5) for clarification.

In connection to the previous point, it is not clear that how does the typical inter-individual distance in the current wind tunnel experiment relate to typical values for free flying groups of zebra finches. Is density in the wind tunnel in the order of natural free flight or rather densely packed?

We are unfortunately not able to answer this question. To the best of our knowledge, there is no data published that describes inter-individual distances of zebra finches in free flight.

The trajectory data analysis is mostly based on detecting the periodic oscillation in the signal by using periodogram function. That should be correct if a periodicity in the signal exists, which is the case for many of the examples given. But if the signal does not have regular changes (because oscillation occurs irregularly or oscillations are not even present), then the values given by that function are rather misleading. The power spectrum will still have a highest value, but reporting that is not very informative. For example, Fig. 1E shows birds with mostly periodic oscillation (horizontal position:

13

23purple, yellow, blue) but also shows birds that have irregular oscillations (horizontal position: magenta, black). The periodicity value given by f does not capture whether the oscillation is present or not. Horizontal position: magenta and vertical position: yellow have almost the same f value, and the former has oscillation with a characteristic time duration just happening at irregular times, and the later does not have significant oscillation. Also, if the signal has lower and higher frequency harmonic components, the current method used (i.e. reporting only the highest value of the periodogram) may not give the correct value that actually characterises the motion pattern. This is clear in some of the cases where high frequency oscillation is present in the signal, but still the reported f is very low. I suggest taking into consideration the strength of the power spectrum as well to characterize whether the periodicity is captured by f or not. Unfortunately, this would mean that all the statistics that use f as the main value may need to be recalculated.

A significance criterium was already included in our analysis of movement oscillations. A peak in the periodogram was only considered to be significant if it exceeded 5 standard deviations of the mean of the whole spectrum. Unfortunately, we failed to explain this in the previous version of the manuscript. Now, the significance criterium is mentioned in the methods section (page 32 lines 8-12). If a periodogram did not possess a significant peak, f was given the value zero in the previous versions of Fig. 1e and S2. To avoid confusion, in the revised version of both figures (now Fig. 1e-g and Extended Data Fig. 2), f is labeled n.s. for movement trajectories without significant oscillations.

The Matlab code used to generate previous Fig. 1E included a mistake when calculating the oscillation frequency for the horizontal movements of bird Pink (previously 0.09 Hz). This was corrected, and f for horizontal movements of bird Pink now assumes a value of 0.47 Hz in Fig. 1e.

For some of the reported statistics, it is not clear what was used as “data points”. There are several places where n is not reported (if there is some common value for all of them that still should be clearly stated somewhere). Were only independent points used for the statistics, so no artificial inflation of number of data points occurs? How was that assured? Could the authors comment on that?

Sample sizes for all statistical tests are now provided in the text.

Please note that we repeated all statistic calculations, but now fitted Linear mixed models to our datasets instead of using Mann-Whitney U-tests, to account for the repeated measures in our experiments. All statistic tests were performed with the statistics toolbox for Matlab. We are not aware of an artificial inflation of the number of data points by the Matlab statistics functions.

The authors use correlation to detect coordinated movement within the flock. The individuals perform motion patterns composed of multiple spatial and temporal scales. It is correct to check correlated movement in the co-moving coordinate system as done by using tracking data in the wind tunnel. But I think the description gives a misleading impression that during most parts of the flight the birds move in an uncorrelated way. In contrast to this, the main flight direction is same for all birds so their motion is very much correlated, as the main direction of the motion is forced by the air flow in the wind tunnel. I suggest to clarify this at around the first paragraph of page 6.

Yes, you are right, the main direction of movement, which is forced by the air flow, is aligned amongst all flock members throughout all flight sessions performed in the wind tunnel. However, this alignment cannot be captured with a video camera because there is no change

of the birds' positions in the direction of the air flow. We clarified this in the text at page 6 lines 4-8.

I really liked the experiments with low light intensity and with additional noise to mask the calls. The results of these experiments strongly indicate the importance of using specific information to avoid collisions.

Thank you.

Minor comments:

Maybe I missed, but I did not find the definition about the meaning of the spatial coordinates. On the figure "arbitrary unit" is being used, I guess that the arbitrary unit relates to the pixel coordinates provided by the video tracking. If so, the values reported are strongly affected by parallax and perspective effects. Although the current analyses (based on power spectrum, for example) may not be affected by that (which is a clever choice of the authors), I think it would necessary to include some explanation for this.

Yes, this is correct. The output of the tracking software was in pixel coordinates, which indeed could have been affected by parallax. Therefore, we decided to use "arb. units" for axes labeling instead. This is now explained in the methods section (page 31 line 24 and page 32 lines 1-2).

Maybe I missed this as well, but I did not find (and I found puzzling) where does the value of 209ms originate from (Page 13 Line 16) in relation to Stack calls.

To analyze call-related movements, we tracked the birds' positions in every fifth frame of the footage, originally sampled with 120 Hz, for five steps starting at the time of call onset. Thus, one tracking step covered 41.666 ms, and five tracking steps consequently 208.333 ms. We decided to ceil this value to 209 ms, to simplify the text. We clarified this in the revised manuscript (page 15 lines 8-9).

Page 13 Line 22: The authors state that based on the non-significant Rayleigh test the distribution is uniform. This is problematic and I suggest rewriting this sentence, as the values which are shown on Fig. 3D are far from being uniformly distributed. The Rayleigh test could show if a circular data has a unimodal peak or not, but here the data is rather inhomogeneous having two peaks on both sides. That could cause the observed non-significant output of the test. So I think there should be a more appropriate method used (for example Hermans-Rasson test?). This is true for the other similar plots, although because of the large number of data point, those could be significant using Rayleigh test even for the bimodal cases (shown on panels Fig. 3F - right side).

Yes, this is true. The distribution does not show a significant directionality, which doesn't necessarily mean that it is uniform. We rewrote the sentence at page 15 line 16. In addition, we performed the Hermans-Rasson test on all circular data and found that none of the data sets is uniformly distributed. However, we still feel that the Rayleigh test is here more appropriate to show significant directionalities in the distributions.

The figures are generally nicely structured and neat. One improvement that I would suggest is that the readability could be a bit better supported by using more panel labels and/or text labels on the panels where there are multiple plots belonging to a single panel label (instead of just writing in the legend left/right and top/bottom).

Thanks. All panels in all figures are now clearly labeled.

Page 15 Line 3: I think the value given there has a typo, as the average of the 4 numbers given in the bracket is 10.25 and not 10.025. Moreover (very minor comment), I doubt that the precision of the value is accurate to the 2nd decimal anyways, so I think it would be more readable to give as 41:540 reporting simply the sum rather than the mean of the values.

Yes, thank you, this was indeed a typo. We corrected the value (page 17 line 4), but decided to stick with the mean ratio instead of the sum.

Fig. S2. (position in wind direction plots, for example the top right group) here several positions are outside of the given range. Is this a results of the plotting, or the tracking is not available for these parts? How do such missing points impact the periodogram analysis?

This is an important point, which we did not pay attention to before. Thank you for mentioning it. For the missing values in the plots of positions in wind direction indeed no tracking data were available because the particular bird was outside the camera's field of view. The periodogram analysis in wind direction is likely affected by the missing values. This is now explained in the manuscript (page 32 lines 9-12), and the data on position changes in wind direction has been removed from the manuscript. Because the birds were in the field of view of Camera 1 for the entire time of all flight sessions, tracking data for the horizontal and vertical dimension is not missing any values.

Fig. S7. It is not clear what is shown on the bottom plot?

The bottom panel in Fig. S7 (now Extended Data Fig. 9) shows the magnitude spectrum of the background noise in the flight section during flight sessions with presentation of band-pass filtered noise. Here, the magnitude of the background noise (black) exceeds the magnitude of the calls (red) for all frequencies. This is now explained in the figure caption. We apologize for missing to explain the purpose of the bottom panel in the initial version of the manuscript.

Table S1 (and Page 5 Line 14 for example) Why is there a need for using a relative score that is divided by the maximum distance? This makes these relative values very sensitive to large outliers.

Due to the parallax effect in the footage, spatial distances between birds of a pair that is flying in the front of the flock (i.e. at a large distance to the camera) generally assumes smaller values than distances between birds of a pair that is flying in the back of the flock (i.e. at a small distance to the camera). To allow for the comparison of spatial distances between different bird pairings, we decided to normalize all distance values of each pair to the maximum distance each pair had assumed in each flight session.

Summary

16

26As a consequence of all the above, I suggest a revision, although I think the changes will not impact the main conclusions of the paper. I would be happy to review the improved version of the manuscript.

Thank you again for the helpful comments. We hope that we were able to clarify all issues in our replies to your comments and in the revised version of the manuscript.Decision Letter, first revision:

6th April 2022

Dear Dr. Hoffmann,

Thank you for submitting your revised manuscript "Vision and vocal communication guide 3-D interindividual spatial coordination in a group of zebra finches during flight in a wind tunnel" (NATECOLEVOL-211115220A). It has now been seen again by the original reviewers and their comments are below. The reviewers find that the paper has improved in revision, and therefore we'll be happy in principle to publish it in Nature Ecology & Evolution, pending minor revisions to comply with our editorial and formatting guidelines.

We are now performing detailed checks on your paper and will send you a checklist detailing our editorial and formatting requirements in about a week. Please do not upload the final materials and make any revisions until you receive this additional information from us. In the meantime you could consider reviewer 2's alternative title suggestion--editorially I think this is a good compromise as I agree the latest title is a bit complicated.

[REDACTED]

Reviewer #1 (Remarks to the Author):

The authors have addressed all my points. I think it's a great study and look forward to seeing it in print.

Reviewer #2 (Remarks to the Author):

Having served as Reviewer 2 on the initial submission, I am very happy with the way the points from the 3 reviewers have been addressed and have no further suggestions of consequence. I give only a few minor observations below that you may wish to act upon. Congratulations on an excellent paper, it was a pleasure to review and I look forward to seeing it published.

Title: I feel that in responding to the request of Reviewer 3 the title has become a little on the heavy side. Might 'Visual and vocal communication guide 3D spatial coordination of zebra finches during

28wind-tunnel flights' be a neater way of putting it?

P3 Line 2: The comma after 'cluster flocks' is not necessary

P3 Line 8: The word indeed could be removed here.

P20 Line 10: Is this the correct test to report here? The paragraphs preceding report LMM and Raleigh tests?

Reviewer #3 (Remarks to the Author):

In the revised version of the manuscript now titled "Vision and vocal communication guide 3-D interindividual spatial coordination in a group of zebra finches during flight in a wind tunnel", the authors made a thorough work with the revision and took into consideration most of the points raised by all reviewers. I accept the answers, and I find that the changes made are satisfactory. After careful examination of the revised manuscript and the response letter, I don't have further comments or suggestions. Thank you for your work. As a consequence, I support acceptance.

Mate Nagy

Our ref: NATECOLEVOL-211115220A

8th April 2022

Dear Dr. Hoffmann,

Thank you for your patience as we've prepared the guidelines for final submission of your Nature Ecology & Evolution manuscript, "Vision and vocal communication guide 3-D interindividual spatial coordination in a group of zebra finches during flight in a wind tunnel" (NATECOLEVOL-211115220A). Please carefully follow the step-by-step instructions provided in the attached file, and add a response in each row of the table to indicate the changes that you have made. Please also check and comment on any additional marked-up edits we have proposed within the text. Ensuring that each point is addressed will help to ensure that your revised manuscript can be swiftly handed over to our production team.

29****We would like to start working on your revised paper, with all of the requested files and forms, as soon as possible (preferably within two weeks). Please get in contact with us immediately if you anticipate it taking more than two weeks to submit these revised files.****

In recognition of the time and expertise our reviewers provide to Nature Ecology & Evolution's editorial process, we would like to formally acknowledge their contribution to the external peer review of your manuscript entitled "Vision and vocal communication guide 3-D interindividual spatial coordination in a group of zebra finches during flight in a wind tunnel". For those reviewers who give their assent, we will be publishing their names alongside the published article.

Nature Ecology & Evolution offers a Transparent Peer Review option for new original research manuscripts submitted after December 1st, 2019. As part of this initiative, we encourage our authors to support increased transparency into the peer review process by agreeing to have the reviewer comments, author rebuttal letters, and editorial decision letters published as a Supplementary item. When you submit your final files please clearly state in your cover letter whether or not you would like to participate in this initiative. Please note that failure to state your preference will result in delays in accepting your manuscript for publication.

Cover suggestions

As you prepare your final files we encourage you to consider whether you have any images or illustrations that may be appropriate for use on the cover of Nature Ecology & Evolution.

30Nature Ecology & Evolution has now transitioned to a unified Rights Collection system which will allow our Author Services team to quickly and easily collect the rights and permissions required to publish your work. Approximately 10 days after your paper is formally accepted, you will receive an email in providing you with a link to complete the grant of rights. If your paper is eligible for Open Access, our Author Services team will also be in touch regarding any additional information that may be required to arrange payment for your article.

Please note that *Nature Ecology & Evolution* is a Transformative Journal (TJ). Authors may publish their research with us through the traditional subscription access route or make their paper immediately open access through payment of an article-processing charge (APC). Authors will not be required to make a final decision about access to their article until it has been accepted. [Find out more about Transformative Journals](https://www.springernature.com/gp/open-research/transformative-journals)

Authors may need to take specific actions to achieve [compliance with funder and institutional open access mandates](https://www.springernature.com/gp/open-research/funding/policy-compliance-faqs). If your research is supported by a funder that requires immediate open access (e.g. according to [Plan S principles](https://www.springernature.com/gp/open-research/plan-s-compliance)) then you should select the gold OA route, and we will direct you to the compliant route where possible. For authors selecting the subscription publication route, the journal's standard licensing terms will need to be accepted, including [a self-archiving and license to publish](https://www.nature.com/nature-portfolio/editorial-policies/self-archiving-and-license-to-publish). Those licensing terms will supersede any other terms that the author or any third party may assert apply to any version of the manuscript.

[REDACTED]

[REDACTED]

Reviewer #1:

Remarks to the Author:

The authors have addressed all my points. I think it's a great study and look forward to seeing it in print.

Reviewer #2:

Remarks to the Author:

Having served as Reviewer 2 on the initial submission, I am very happy with the way the points from the 3 reviewers have been addressed and have no further suggestions of consequence. I give only a few minor observations below that you may wish to act upon. Congratulations on an excellent paper, it was a pleasure to review and I look forward to seeing it published.

Title: I feel that in responding to the request of Reviewer 3 the title has become a little on the heavy side. Might 'Visual and vocal communication guide 3D spatial coordination of zebra finches during wind-tunnel flights' be a neater way of putting it?

P3 Line 2: The comma after 'cluster flocks' is not necessary

P3 Line 8: The word indeed could be removed here.

P20 Line 10: Is this the correct test to report here? The paragraphs preceding report LMM and Raleigh tests?

Reviewer #3:

Remarks to the Author:

In the revised version of the manuscript now titled "Vision and vocal communication guide 3-D interindividual spatial coordination in a group of zebra finches during flight in a wind tunnel", the authors made a thorough work with the revision and took into consideration most of the points raised by all reviewers. I accept the answers, and I find that the changes made are satisfactory. After careful examination of the revised manuscript and the response letter, I don't have further comments or suggestions. Thank you for your work. As a consequence, I support acceptance.

Mate Nagy

Author Rebuttal, first revision

Reviewer #1:

The authors have addressed all my points. I think it's a great study and look forward to seeing it in print.

We thank the reviewer for the time invested to review and to improve our manuscript.

Max-Planck-Institut für Ornithologie
Eberhard-Gwinner-Straße · 82319 Seewiesen · Germany
Tel.: +49 (0) 8157-932-0 · Fax: +49 (0) 8157-932-400 · www.orn.mpg.de

Max-Planck-Institut für Ornithologie · Vogelwarte Radolfzell
Am Obstberg 1 · 78315 Radolfzell · Germany
Tel.: +49 (0) 7732-1501-0 · Fax: +49 (0) 7732-1501-69 · www.orn.mpg.de

MAX-PLANCK-GESellschaftReviewer #2:

Having served as Reviewer 2 on the initial submission, I am very happy with the way the points from the 3 reviewers have been addressed and have no further suggestions of consequence. I give only a few minor observations below that you may wish to act upon. Congratulations on an excellent paper, it was a pleasure to review and I look forward to seeing it published.

We thank the reviewer for the time invested to review and to improve our manuscript.

Title: I feel that in responding to the request of Reviewer 3 the title has become a little on the heavy side. Might 'Visual and vocal communication guide 3D spatial coordination of zebra finches during wind-tunnel flights' be a neater way of putting it?

Thank you, we agree and changed the title accordingly.

P3 Line 2: The comma after 'cluster flocks' is not necessary

We removed the comma.

P3 Line 8: The word indeed could be removed here.

The word "indeed" was removed.

P20 Line 10: Is this the correct test to report here? The paragraphs preceding report LMM and Raleigh tests?

The correct test is now reported in the figure legend.

Reviewer #3:

In the revised version of the manuscript now titled "Vision and vocal communication guide 3-D interindividual spatial coordination in a group of zebra finches during flight in a wind tunnel", the authors made a thorough work with the revision and took into consideration most of the points raised by all reviewers. I accept the answers, and I find that the changes made are satisfactory. After careful examination of the revised manuscript and the response letter, I don't have further comments or suggestions. Thank you for your work. As a consequence, I support acceptance.

Mate Nagy

Thank you, Dr. Nagy, for reviewing our manuscript and for your help to improve its quality.

Final Decision Letter:

19th May 2022

Dear Dr Hoffmann,

We are pleased to inform you that your Article entitled "Vision and vocal communication guide 3D spatial coordination of zebra finches during wind-tunnel flights", has now been accepted for publication in Nature Ecology & Evolution.

Over the next few weeks, your paper will be copyedited to ensure that it conforms to Nature Ecology and Evolution style. Once your paper is typeset, you will receive an email with a link to choose the appropriate publishing options for your paper and our Author Services team will be in touch regarding any additional information that may be required

You will not receive your proofs until the publishing agreement has been received through our system

Due to the importance of these deadlines, we ask you please us know now whether you will be difficult to contact over the next month. If this is the case, we ask you provide us with the contact information (email, phone and fax) of someone who will be able to check the proofs on your behalf, and who will be available to address any last-minute problems . Once your paper has been scheduled for online publication, the Nature press office will be in touch to confirm the details.

Acceptance of your manuscript is conditional on all authors' agreement with our publication policies (see www.nature.com/authors/policies/index.html). In particular your manuscript must not be published elsewhere and there must be no announcement of the work to any media outlet until the publication date (the day on which it is uploaded onto our web site).

Please note that *Nature Ecology & Evolution* is a Transformative Journal (TJ). Authors may publish their research with us through the traditional subscription access route or make their paper immediately open access through payment of an article-processing charge (APC). Authors will not be required to make a final decision about access to their article until it has been accepted. [Find out more about Transformative Journals](https://www.springernature.com/gp/open-research/transformative-journals)

Authors may need to take specific actions to achieve [compliance with funder and institutional open access mandates](https://www.springernature.com/gp/open-research/funding/policy-compliance-faqs). If your research is supported by a funder that requires immediate open access (e.g. according to <https://www.springernature.com/gp/open-research/funding/policy-compliance-faqs>)

35[Plan S principles](https://www.springernature.com/gp/open-research/plan-s-compliance)) then you should select the gold OA route, and we will direct you to the compliant route where possible. For authors selecting the subscription publication route, the journal's standard licensing terms will need to be accepted, including <https://www.nature.com/nature-portfolio/editorial-policies/self-archiving-and-license-to-publish>. Those licensing terms will supersede any other terms that the author or any third party may assert apply to any version of the manuscript.

We welcome the submission of potential cover material (including a short caption of around 40 words) related to your manuscript; suggestions should be sent to Nature Ecology & Evolution as electronic files (the image should be 300 dpi at 210 x 297 mm in either TIFF or JPEG format). Please note that such pictures should be selected more for their aesthetic appeal than for their scientific content, and that colour images work better than black and white or grayscale images. Please do not try to design a cover with the Nature Ecology & Evolution logo etc., and please do not submit composites of images related to your work. I am sure you will understand that we cannot make any promise as to whether any of your suggestions might be selected for the cover of the journal.

You can generate the link yourself when you receive your article DOI by entering it here: <http://authors.springernature.com/share>.

[REDACTED]

P.S. Click on the following link if you would like to recommend Nature Ecology & Evolution to your librarian <http://www.nature.com/subscriptions/recommend.html#forms>

** Visit the Springer Nature Editorial and Publishing website at http://editorial-jobs.springernature.com?utm_source=ejp_NEcoE_email&utm_medium=ejp_NEcoE_email&utm_campaign=ejp_NEcoE for more information about our career opportunities. If you have any questions please click [here](mailto:editorial.publishing.jobs@springernature.com).**